# Attacks on Online Learners: a Teacher-Student Analysis

**Riccardo G. Margiotta**[*]     **Sebastian Goldt**     **Guido Sanguinetti**

*International School for Advanced Studies, Trieste, Italy*

## Abstract

Machine learning models are famously vulnerable to adversarial attacks: small ad-hoc perturbations of the data that can catastrophically alter the model predictions. While a large literature has studied the case of test-time attacks on pre-trained models, the important case of attacks in an online learning setting has received little attention so far. In this work, we use a control-theoretical perspective to study the scenario where an attacker may perturb data labels to manipulate the learning dynamics of an online learner. We perform a theoretical analysis of the problem in a teacher-student setup, considering different attack strategies, and obtaining analytical results for the steady state of simple linear learners. These results enable us to prove that a discontinuous transition in the learner's accuracy occurs when the attack strength exceeds a critical threshold. We then study empirically attacks on learners with complex architectures using real data, confirming the insights of our theoretical analysis. Our findings show that greedy attacks can be extremely efficient, especially when data stream in small batches.

## 1 Introduction

Adversarial attacks pose a remarkable threat to modern machine learning (ML), as models based on deep networks have proven highly vulnerable to such attacks. Understanding adversarial attacks is therefore of paramount importance, and much work has been done in this direction; see [1, 2] for reviews. The standard setup of adversarial attacks aims to change the predictions of a trained model by applying a minimal perturbation to its inputs. In the *data poisoning* scenario, instead, inputs and/or outputs of the training data are corrupted by an attacker whose goal is to force the learner to get as close as possible to a "nefarious" target model, for example to include a backdoor [3, 4]. Data poisoning has also received considerable attention that has focused on the offline setting, where models are trained on fixed datasets [5]. An increasing number of real-world applications instead require that machine learning models are continuously updated using streams of data, often relying on transfer-learning techniques. In many cases, the streaming data can be subject to adversarial manipulations. Examples include learning from user-generated data (GPT-based models) and collecting information from the environment, as in e-commerce applications. Federated learning constitutes yet another important example where machine learning models are updated over time, and where malicious nodes in the federation have private access to a subset of the data used for training.

In the online setting, the attacker intercepts the data stream and modifies it to move the learner towards the nefarious target model, see Fig. 1-A for a schematic where attacks change the labels. In this streaming scenario, the attacker needs to predict both the future data stream and model states, accounting for the effect of its own perturbations, so as to decide how to poison the current data batch. Finding the best possible attack policy for a given data batch and learner state can be formalized as a stochastic optimal control problem, with opposing costs given by the magnitude of perturbations

---

[*]R. G. Margiotta (`rimargi@sissa.it`) is the corresponding author.

37th Conference on Neural Information Processing Systems (NeurIPS 2023).

and the distance between learner and attacker's target [6]. The ensuing dynamics have surprising features. Take for example Fig. 1-B, where we show the accuracy of an attacked model (a logistic regression classifier classifying MNIST images) as a function of the attack strength (we define this quantity in Sec. 2.2). Even in such a simple model, we observe a strong nonlinear behavior where the accuracy drops dramatically when crossing a critical attack strength. This observation raises several questions about the vulnerability of learning algorithms to online data poisoning. What is the effect of the perturbations on the learning dynamics and long-term behavior of the model under attack? Is there a minimum perturbation strength required for the attacker to achieve a predetermined goal? And how do different attack strategies compare in terms of efficacy? We investigate these questions by analyzing the attacker's control problem from a theoretical standpoint in the teacher-student setup, a popular framework for studying machine learning models in a controlled way [7–10]. We obtain analytical results characterizing the steady state of the attacked model in a linear regression setting, and we show that a simple greedy attack strategy can perform near optimally. This observation is empirically replicated across more complex learners in the teacher-student setup, and the fundamental aspects of our theoretical results are also reflected in real-data experiments.

**Main contributions**

- We present a theoretical analysis of online data poisoning in the teacher-student setup, providing analytical results for the linear regression case [RESULT 1-5]. In particular, we demonstrate that a phase transition takes place when the batch size approaches infinity, signaled by a discontinuity in the accuracy of the student against the attacker's strength [RESULT 2].

- We provide a quantitative comparison between different attack strategies. Surprisingly, we find that properly calibrated *greedy* attacks can be as effective as attacks with full knowledge of the incoming data stream. Greedy attacks are also computationally efficient, thus constituting a remarkable threat to online learners.

- We empirically study online data poisoning on real datasets (MNIST, CIFAR10), using architectures of varying complexities including LeNet, ResNet, and VGG. We observe qualitative features that are consistent with our theoretical predictions, providing validation for our findings.

## 1.1 Further related work

Data poisoning attacks have been the subject of a wide range of studies. Most of this research focused on the offline (or batch) setting, where the attacker interacts with the data only once before training starts [3, 11–14]. Mei et al. notably proposed an optimization approach for offline data poisoning [15]. Wang and Chaudhuri addressed a streaming-data setting, but in the idealized scenario where the attacker has access to the full (future) data stream (the so-called *clairvoyant* setting), which effectively reduces the online problem to an offline optimization [16]. So far, the case of online data poisoning has received only a few contributions with a genuine online perspective. In [17], Pang and coauthors investigated attacks on online learners with poisoned batches that aim for an immediate disruptive impact rather than manipulating the long-term learning dynamics. Zhang et al. proposed an optimal control formulation of online data poisoning by following a parallel line of research, that of online teaching [18, 19], focussing then on practical attack strategies that perturb the input [6]. Our optimal control perspective is similar to that of Zhang and co-authors, though our contribution differs from theirs in various aspects. First, we consider a supervised learning setting where attacks involve data labels, and not input data. We note that attacks on the labels have a reduced computational cost compared to high-dimensional adversarial perturbations of the inputs, which makes label poisoning more convenient in a streaming scenario. Second, we provide a theoretical analysis and explicit results for the dynamics of the attacked models, and we consider non-linear predictors in our experiments. We observe that our setup is reminiscent of continual learning, which has also recently been analyzed in a teacher-student setup [20]. However, in online data poisoning the labels are dynamically changed by the attacker to drive the learner towards a desired target. In continual learning there is no entity that actively modifies the labels, and the learner simply switches from one task to another. Finally, we note that standard (static, test-time) adversarial attacks have been studied in a teacher-student framework [21].

**Reproducibility**. The code and details for implementing our experiments are available here.

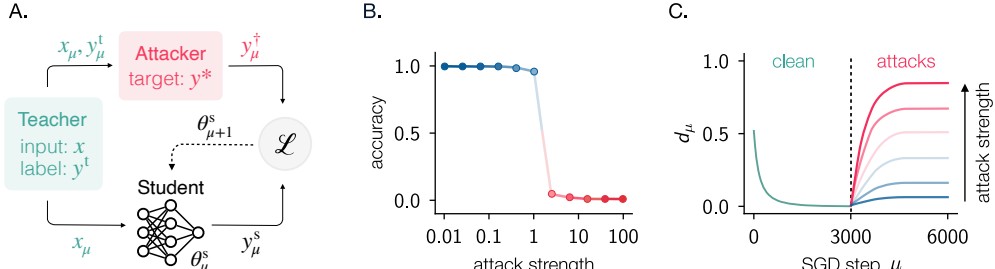

Figure 1: *Problem setup and key experimental observations*. A: The environment (teacher) generates a sequence of clean data with $\mu^{\text{th}}$ input $x_\mu$ and label $y_\mu^{\text{t}}$. The attacker perturbs the labels towards a nefarious target $y_\mu^* = \phi^*(x_\mu)$. At each training step, the predictions $y_\mu^{\text{s}}$ of the learner (student) and the perturbed labels $y_\mu^\dagger$ are used to compute the loss and update the learner parameters. B: accuracy vs attack strength of a logistic regression classifying MNIST digits 1 and 7 under online label-flipping attacks. C: relative distance $d_\mu$ of the logistic learner vs SGD step $\mu$ during the clean training phase, and for training under attacks of increasing attack strength.

## 2 Problem setup

### 2.1 The teacher, the student, and the attacker

The problem setup is depicted in Fig. 1-A. There are three entities: the teacher, the student, and the attacker. The *teacher* is a model of the environment, which generates a stream of clean input-output data batches. The inputs $x \in \mathbb{R}^D$ have elements sampled from a distribution $\mathcal{P}_x$, while clean labels are provided by the scalar teacher function: $y^{\text{t}} = \phi^{\text{t}}(x)$. The $\mu^{\text{th}}$ batch of the stream is $B_\mu^P = \{(x_\mu, y_\mu^{\text{t}})_p\}_{p=1}^P$, with $P$ the batch size. The *attacker* intercepts the clean batch $B_\mu^P$ and perturbs the labels so as to make the student learn the *target* function $\phi^*$. The *student* $\phi_\mu^{\text{s}}$ represents the learning model, and it updates its weights $\theta_\mu^{\text{s}}$ on the stream of perturbed batches via SGD, using each (perturbed) batch only once. We will consider teacher, attacker's target, and student with the same architecture $\phi(x; \theta)$; they only differ by the value of their parameters, respectively $\theta^{\text{t}}, \theta^*, \theta_\mu^{\text{s}}$. We will refer to this Teacher-Student-Attacker setup as the *TSA problem*.

If the student trained directly on the input-label pairs provided by the teacher, we would recover the standard teacher-student setup that has been analyzed extensively in the literature [7–10]. The key novelty of this paper is that we consider a setup where each new batch of data is instead *poisoned* by the attacker with the attempt of deviating the student towards a different target function.

### 2.2 The attack

The goal of the attacker is for the student to predict a target label $y^* = \phi^*(x)$ for each given input $x$. Target label and clean label can coincide for specific inputs, however they won't generally do so. The attacker performs an action $a_\mu$ that produces perturbed labels as

$$y_\mu^\dagger = y^{\text{t}}(1 - a_\mu) + y^* a_\mu, \quad a_\mu \in [a_{\min}, a_{\max}]. \tag{1}$$

Note that $a_\mu = 0$ implies no perturbation, while for $a_\mu = 1$ perturbed and target labels coincide. We allow $a_{\min} < 0$ and $a_{\max} > 1$ to model attackers that under/overshoot. The attacker's action $a_\mu$ thus constitutes the *control variable* of the TSA problem. We primarily consider the case where the same value of $a_\mu$ is used for all labels in each batch, but can be different for different batches. We also explore settings where this assumption is relaxed, including the case of sample-specific perturbations, and the scenario where only a fraction of the labels in each batch is perturbed.

The aim of the attacker is to sway the student towards its target, and so minimize the *nefarious cost*

$$g_\mu^{\text{nef}} = \frac{1}{2P} \sum_{p=1}^P (\phi_\mu^{\text{s}}(x_{\mu p}) - \phi^*(x_{\mu p}))^2, \tag{2}$$

using small perturbations. For this reason, it also tries to minimize the *perturbation cost*

$$g_\mu^{\mathrm{per}} = \frac{1}{2}\tilde{C}a_\mu^2, \qquad \tilde{C} = \mathcal{E}(\phi^*)C, \tag{3}$$

with $C$ a parameter expressing the cost of actions, and $\mathcal{E}(\phi^*) = \mathbb{E}_x\left[(\phi^*(x) - \phi^{\mathrm{t}}(x))^2\right]$ the expected squared error of the attacker's target function; $\mathbb{E}_x$ indicates the average over the input distribution. We use the pre-factor $\mathcal{E}(\phi^*)$ in $g_\mu^{\mathrm{per}}$ to set aside the effect of the teacher-target distance on the balance between the two costs. Note that $C$ is inversely proportional to the intuitive notion of *attack strength*: low $C$ implies a low cost for the action and therefore results in larger perturbations; the $x$-axis in Fig. 1-B shows the inverse of $C$. The attacker's goal is to find the *sequence* of actions $a_\mu^{\mathrm{opt}}$ that minimize the *total expected cost*:

$$\{a_\mu^{\mathrm{opt}}\} = \operatorname*{argmin}_{\{a_\mu\}} \mathbb{E}_{\mathrm{fut}}\left[\lim_{T \to \infty} G(\{a_\mu\}, T)\right], \qquad G = \sum_{\mu=1}^{T} \gamma^\mu(g_\mu^{\mathrm{per}} + g_\mu^{\mathrm{nef}}), \tag{4}$$

where $\mathbb{E}_{\mathrm{fut}}$ represents the average over all possible realizations of the future data stream, and $\gamma \in (0, 1)$ is the future discounting factor. In relation to $\gamma$, we make the crucial assumption that $G$ is dominated by the steady-state running costs; we use $\gamma = 0.995$ in our experiments.

## 2.3 SGD dynamics under attack

The perturbed batch $B_\mu^{P\dagger} = \{(x_\mu, y_\mu^\dagger)_p\}_{p=1}^P$ is used to update the model parameters $\theta_\mu^{\mathrm{s}}$ following the standard (stochastic) gradient descent algorithm using the MSE loss function:

$$\theta_{\mu+1}^{\mathrm{s}} = \theta_\mu^{\mathrm{s}} - \eta \nabla_{\theta_\mu^{\mathrm{s}}} \mathcal{L}_\mu\left(B_\mu^{P\dagger}\right), \qquad \mathcal{L}_\mu = \frac{1}{2P} \sum_{p=1}^{P} (\phi_\mu^{\mathrm{s}}(x_{\mu p}) - y_{\mu p}^\dagger)^2, \tag{5}$$

where $\eta$ is the learning rate. We characterize the system's dynamics in terms of the relative teacher-student distance, defined as

$$d_\mu(C, P) = \left(\mathcal{E}(\phi_\mu^{\mathrm{s}})/\mathcal{E}(\phi^*)\right)^{1/2}. \tag{6}$$

Note that $d = 0$ (resp. $d = 1$) for a student $\phi^{\mathrm{s}}$ that perfectly predicts the clean (resp. target) labels, on average. We will split the training process into two phases: first, the student trains on clean data, while the attacker calibrates its policy. Then, after convergence of the student, attacks start and training continues on perturbed data. A typical realization of the dynamics is depicted in Fig. 1-C, which shows $d_\mu$ for a logistic regression classifying MNIST digits during the clean-labels phase, and under attacks of increasing strength (decreasing values of the cost $C$). The sharp transition observed in Fig. 1-B occurs for $d_{\mu \to \infty} \approx 0.5$, when the student is equally distant from the teacher and target.

## 2.4 Attack strategies

Finding the optimal actions sequence that solves the control problem (4) involves performing the average $\mathbb{E}_{\mathrm{fut}}$, and so it is only possible if the attacker knows the data generating process. Even then, however, optimal solutions can be found only for simple cases, and realistic settings require approximate solutions. We consider the following approaches:

- **Constant attacks**. The action is observation-independent and remains constant for all SGD steps: $a_\mu = a^{\mathrm{c}} \forall \mu$. This simple baseline can be optimized by sampling data streams using past observations, simulating the dynamics via Eq. (5), and obtaining numerically the value of $a^{\mathrm{c}}$ that minimizes the simulated total expected cost.

- **Greedy attacks**. A greedy attacker maximizes the immediate reward. Since the attacker actions have no instantaneous effect on the learner, the greedy objective has to include the next training step. The greedy action is then given by

$$a_\mu^{\mathrm{G}}(x_\mu, \theta_\mu^{\mathrm{s}}; \tilde{\gamma}) = \operatorname*{argmin}_{a_\mu} \mathbb{E}_{x_{\mu+1}}\left[g_\mu^{\mathrm{per}} + \tilde{\gamma}g_{\mu+1}^{\mathrm{nef}}\right]. \tag{7}$$

The average $\mathbb{E}_{x_{\mu+1}}$ can be estimated with input data sampled from past observations, and using (5) to obtain the next student state. Similarly to constant attacks, the greedy future weight $\tilde{\gamma}$ can be calibrated by minimizing the simulated total expected cost using past observations.

- **Reinforcement learning attacks**. The attacker can use a parametrized policy function $a(x, \theta^{\mathrm{s}})$ and optimize it via reinforcement learning (RL). This is a *model free* approach, and it is suitable for *grey box* settings. Tuning deep policies is computationally costly, however, and it requires careful calibrations. We utilize TD3 [22], a powerful RL agent that improves the deep deterministic policy gradient algorithm [23]. We employ TD3 as implemented in Stable Baselines3 [24].

- **Clairvoyant attacks**. A clairvoyant attacker has full knowledge of the future data stream, so it simply looks for the sequence $\{a_\mu\}$ that minimizes $G(\{a_\mu\}, T \gg 1)$. Although the clairvoyant setting is not realistically plausible, it provides an upper bound for the attacker's performance in our experiments. We use Opty [25] to cast the clairvoyant problem into a standard nonlinear program, which is then solved by an interior-point solver (IPOPT) [26].

The above attack strategies are summarized as pseudocode in Appendix A.

## 3 Theoretical analysis

We first analyze online label attacks with synthetic data to establish our theoretical results. We will confirm our findings with real data experiments in Sec. 4.2. Solving the stochastic optimal control problem (4) is very challenging, even for simple architectures $\phi(x, \theta)$ and assuming that $\mathcal{P}_x$ is known. We discuss an analytical treatment in two scenarios for the *linear* TSA problem with unbounded action space, where

$$\phi(x; w) = w^{\mathrm{T}} x / \sqrt{D}. \tag{8}$$

### 3.1 Large batch size: deterministic limit

In the limit of very large batches, batch averages of the nefarious cost $g_\mu^{\mathrm{nef}}$ and loss $\mathcal{L}_\mu$ in Eqs. (2, 5) can be approximated with averages over the input distribution. For standardized and i.i.d. input elements, the resulting equations are

$$g_\mu^{\mathrm{nef}} = \frac{1}{2D} |w_\mu^{\mathrm{s}} - w^*|^2, \quad \nabla_{w^{\mathrm{s}}} \mathcal{L}_\mu = \frac{1}{D} \left( (w_\mu^{\mathrm{s}} - w^{\mathrm{t}}) + (w_\mu^{\mathrm{t}} - w^*) a_\mu \right). \tag{9}$$

In this setting, the attacker faces a deterministic optimal control problem, which can be solved using a Lagrangian-based method. Following this procedure, we obtain explicit solutions for the steady-state student weights $w_\infty^{\mathrm{s}}$ and optimal action $a_\infty^{\mathrm{opt}}$. The relative distance $d_\infty^{\mathrm{opt}}$ between student and teacher then follows from Eq. (6). We find

$$[\text{RESULT 1}] \qquad w_\infty^{\mathrm{s}} = w^{\mathrm{t}} + d_\infty^{\mathrm{opt}} (w^* - w^{\mathrm{t}}), \qquad d_\infty^{\mathrm{opt}} = a_\infty^{\mathrm{opt}} = (C + 1)^{-1}. \tag{10}$$

The above result is intuitive: the student learns a function that approaches the attacker's target as the cost of actions $C$ tends to zero, and $d_\infty^{\mathrm{opt}}(C = 0) = 1$. Vice versa, for increasing values of $C$, the attacker's optimal action decreases, and the student approaches the teacher function.

In the context of a classification task, we can characterize the system dynamics in terms of the accuracy of the student $A_\mu(C, P) = \mathbb{E}_x[S_\mu(x)]$, with $S_\mu(x) = \left( \mathrm{sign}(\phi_\mu^{\mathrm{s}}(x)) + \mathrm{sign}(\phi^{\mathrm{t}}(x)) \right) / 2$. For label-flipping attacks, where $\phi^* = -\phi^{\mathrm{t}}$, a direct consequence of Eq. (10) is that the steady-state accuracy of the student exhibits a discontinuous transition in $C = 1$. More precisely, we find

$$[\text{RESULT 2}] \qquad\qquad A_\infty(C) = 1 - H(C - 1), \tag{11}$$

with $H(\cdot)$ the Heaviside step function. This simple result explains the behaviour observed in Fig. 1-B, where the collapse in accuracy occurs for attack strength $C^{-1} \approx 1$. We refer to Appendix B for a detailed derivation of results (10, 11).

### 3.2 Finite batch size: greedy attacks

When dealing with batches of finite size, the attacker's optimal control problem is stochastic and has no straightforward solution, even for the simple linear TSA problem. To make progress, we consider greedy attacks, which are computationally convenient and analytically tractable. While an explicit solution to (7) is generally hard to find, it is possible for the linear case of Eq. (8).

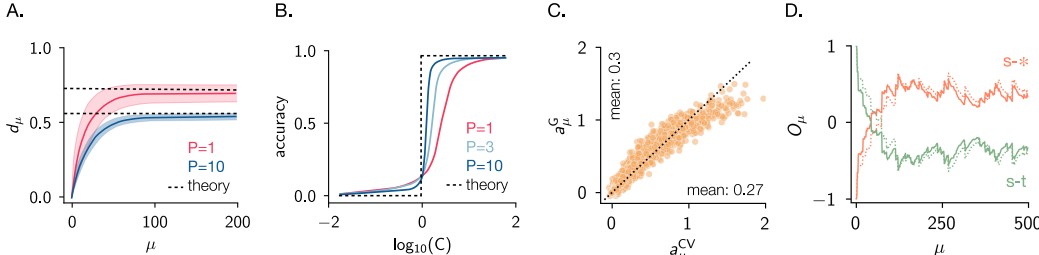

Figure 2: *Greedy and clairvoyant attacks in the linear TSA problem*. A: relative distance vs SGD step $\mu$ for greedy attacks on linear models for two different batch sizes $P$. Solid lines display the average $\bar{d}_\mu$, shaded areas cover 1std of $d_\mu$. Dashed lines show the theoretical estimate of Eqs. (13). B: steady-state average accuracy of the student vs cost of action. The dashed line shows the theoretical prediction of Eq. (11). C: scatterplot of greedy (G) vs clairvoyant (CV) actions used on the same data stream with $P = 1$. D: student-teacher (s-t, green) and student-target (s-*, coral) overlap vs SGD step, for the clairvoyant (dotted line) and greedy (continuous line) attacks of panel C. Parameters: $C = 1$, $a \in [-2, 3]$, $D = 10$, $\eta = 0.02 \times D$. Input elements sampled i.i.d. from $\mathcal{P}_x = \mathcal{N}(0, 1)$.

We find the *greedy action policy*

$$a^{\mathrm{G}} = \frac{1}{\tilde{C}DP} \sum_{p=1}^{P} (w^{\mathrm{s}} - w^*)^{\mathrm{T}} x_p (w^{\mathrm{t}} - w^*)^{\mathrm{T}} x_p + O(\eta). \tag{12}$$

Note that $a^{\mathrm{G}}$ decreases with the cost of actions, and $a^{\mathrm{G}} \to 0$ for $\tilde{C} \to \infty$. With a policy function at hand, we can easily investigate the steady state of the system by averaging over multiple realizations of the data stream. For input elements sampled i.i.d. from $\mathcal{P}_x = \mathcal{N}(0, 1)$, we find the *average* steady-state weights $\bar{w}^{\mathrm{s}}_\infty$ as

[RESULT 3] $\quad\quad \bar{w}^{\mathrm{s}}_\infty = w^{\mathrm{t}} + \bar{d}^{\mathrm{G}}_\infty (w^* - w^{\mathrm{t}}), \quad \bar{d}^{\mathrm{G}}_\infty = \left( \left( \frac{P}{P+2} \right) C + 1 \right)^{-1}, \quad\quad$ (13)

where $\bar{d}^{\mathrm{G}}_\infty$ is the distance (6) of the average steady-state student; we refer to Appendix C for the derivation of the above result. The expression of $\bar{d}^{\mathrm{G}}_\infty$ includes a surprising batch-size effect: for a fixed value of the cost $C$, greedy attacks become more effective for batches of decreasing sizes, as the weights get closer to $w^*$ while perturbations become smaller (see Eq. (45)). This effect is displayed in Fig. 2-A, which shows $d_\mu$ vs SGD step for greedy attacks, and for $P = 1, 10$. Note that only the perturbed training phase is shown, so $d_0 = 0$ as the student converges to the teacher function during the preceding clean training phase. The black dashed lines correspond to the steady-state values given by (13), which are in very good agreement with the empirical averages (up to a finite-$\eta$ effect)[2]. We also observe that $d_\mu$ is characterized by fluctuations with amplitude that decreases as the batch size increases. A consequence of such fluctuations is that the average steady-state accuracy of the student undergoes a smooth transition. The transition becomes sharper as the batch size increases, and in the limit $P \to \infty$ it converges to the prediction (11); see Fig. 2-B, which shows numerical evaluations of $A_\infty(C, P)$ averaged over multiple data streams. In our experiments, we draw the teacher weights elements as i.i.d. samples from $\mathcal{N}(0, 1)$ and normalize them so that $|w^{\mathrm{t}}|^2 = D$, while $w^* = -w^{\mathrm{t}}$.

**Remark.** In order to derive the result of Eq. (13), we have set $\tilde{\gamma} = D/\eta$ so that $\bar{d}^{\mathrm{G}}_\infty \to d^{\mathrm{opt}}_\infty$ for $P \to \infty$ (see Appendix B.1). This guarantees the steady-state optimality of the greedy action policy in the large batch limit. Note that $\tilde{\gamma}$ coincides with the timescale of the linear TSA problem, and it lends itself to a nice interpretation: in the greedy setting, where the horizon ends after the second training step, the future is optimally weighted by a factor proportional to the decorrelation time of the weights. Optimality for $P$ finite is not guaranteed, though we observe numerically that $\tilde{\gamma}$ approximately matches our choice even for $P = 1, 10$. We use again $\tilde{\gamma} = D/\eta$ to obtain the theoretical results of the next two sections.

---

[2]We shall remark that the quantities $\bar{d}^{\mathrm{G}}_\infty$ and $\bar{d}_\infty = \lim_{\mu \to \infty} \langle d_\mu \rangle$, the average over multiple data streams shown in Fig. 2-A, can, in general, differ, and they only coincide necessarily in the deterministic limit $P \to \infty$.

### 3.2.1 Sample-specific perturbations

The batch-size effect observed for greedy attacks can be explained as a signature of the batch average appearing in the policy function (12). When dealing with a single data point at a time, the attacker has precise control over the data labels, designing sample-specific perturbations. For batches with multiple samples, instead, perturbations are designed to perform well on average and do not exploit the fluctuations within the data stream. As a result, attacks become less effective, increasing in magnitude and leading to a smaller average steady-state distance between the student and the teacher function. A straightforward solution to this problem consists of applying sample-specific perturbations, thus using a multi-dimensional control $a \in \mathbb{R}^P$, which, however, results in a higher computational cost. This can be achieved by using the policy function (12) independently for each sample, so that the $p$-th element of $a$ is given by

$$a_p^{\mathrm{G}} = \frac{1}{\tilde{C}D}(w^{\mathrm{s}} - w^*)^{\mathrm{T}}x_p(w^{\mathrm{t}} - w^*)^{\mathrm{T}}x_p. \tag{14}$$

In Appendix C.1, we show that this strategy coincides with the optimal greedy solution for multi-dimensional, sample-specific control, up to corrections of order $\eta$. We also show that the resulting average steady state reached by the student, for normal and i.i.d. input data, is

[RESULT 4] $\qquad \bar{w}_\infty^{\mathrm{s}} = w^{\mathrm{t}} - \bar{d}_\infty^{\mathrm{G}}\Delta w^{\mathrm{t}*}, \qquad \bar{d}_\infty^{\mathrm{G}} = (C/3 + 1)^{-1}. \tag{15}$

Remarkably, there is no batch-size dependence in the above solution, and the average steady-state distance coincides with (13) for $P = 1$. This result demonstrates that precise, sample-specific greedy attacks remain effective independently of the batch size of the streaming data.

### 3.2.2 Mixing clean and perturbed data

In the previous section, we addressed a case of *enhanced* control, where batch perturbations are replaced by sample-specific ones. Here, instead, we consider a case of *reduced* control, where the attacker can apply the same perturbation to a fraction of the samples in each batch. This scenario can arise when the attacker faces computational limitations or environmental constraints. For example, the streaming frequency may be too high for the attacker to contaminate all samples at each time step, or, in a poisoned federated learning setting, the central server could have access to a source of clean data, beyond the reach of the attackers. Concretely, we assume that the attacker has access to a fraction $\rho$ of the batch samples only. Therefore, training involves $\rho P$ perturbed and $(1 - \rho)P$ clean samples. Perturbations follow the policy function

$$a_\mu^{\mathrm{G}} = \frac{1}{\tilde{C}D\rho P}\sum_{p=1}^{\rho P}\Delta w_\mu^{\mathrm{s}*\,\mathrm{T}}x_{\mu p}\Delta w^{\mathrm{t}*\,\mathrm{T}}x_{\mu p} + O(\eta). \tag{16}$$

In Appendix C.2, we find the following average steady-state solution for normal and i.i.d. input data:

[RESULT 5] $\qquad \bar{w}_\infty^{\mathrm{s}} = w^{\mathrm{t}} - \bar{d}_\infty^{\mathrm{G}}\Delta w^{\mathrm{t}*}, \qquad \bar{d}_\infty^{\mathrm{G}} = \left(\left(\frac{P}{\rho P + 2}\right)C + 1\right)^{-1}; \tag{17}$

see Fig. 7 for a comparison with empirical evaluations. Note that for large batches $\bar{d}_\infty^{\mathrm{G}} \simeq (C/\rho + 1)^{-1}$, indicating that the cost of actions effectively increases by a factor $\rho^{-1}$. We remark that the above result is derived under the assumption that the attacker ignores the number of clean samples used during training, so it cannot compensate for the reduced effectiveness of the attacks.

## 4 Empirical results

### 4.1 Experiments on synthetic data

Having derived analytical results for the infinite batch and finite batch linear TSA problem, we now present empirical evidence as to how different strategies compare on different architectures. As a first question, we investigate the efficacy of greedy attacks for data streaming in batches of small size. By choosing an appropriate future weight $\tilde{\gamma}$, we know the greedy solution is optimal as the batch size tends to infinity. To investigate optimality in the finite batch setting, we compare greedy actions with clairvoyant actions on the same data stream, optimizing $\tilde{\gamma}$ over a grid of possible values. By

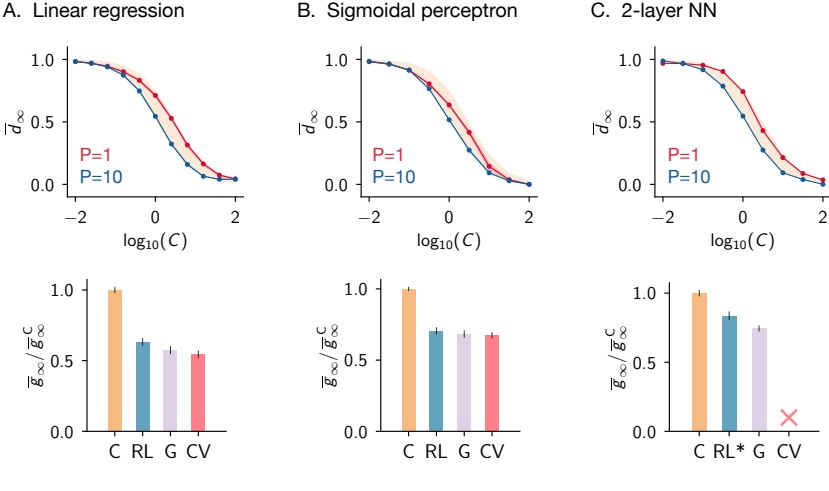

C: constant,  RL: reinforcement learning,  G: greedy,  CV: clairvoyant

Figure 3: *Empirical results for the TSA problem with synthetic data*. A, top panel: average steady-state distance vs cost of action $C$ for the linear TSA problem and greedy attacks. The orange area represents the range of solutions of $\bar{d}_\infty^G$ (13) for $P \in [1, 10]$. Bottom panel: average steady-state running cost of different attack strategies relative to the largest one (constant attacks), for $P = 1$, and $C = 1$. B, C: same quantities as shown in A for the perceptron and NN architectures; RL* indicates an agent that sees the final layer only. Averages were performed over 10 data streams of $10^5$ batches and over the last $10^3$ steps for each stream. Parameters: $D = 10$, $\eta = 0.02 \times D$, $a \in [-2, 3]$ (A, B), $M = 100$, $\eta = 0.02 \times D \times \sqrt{M}$, $a \in [0, 1]$ (C). Input elements sampled i.i.d. from $\mathcal{P}_x = \mathcal{N}(0, 1)$.

definition, clairvoyant attacks have access to all the future information and therefore produce the best actions sequence. The scatterplot in Fig. 2-C shows greedy vs clairvoyant actions, respectively $a^G$ and $a^{CV}$, on the same stream. Note that greedy actions are typically larger than their clairvoyant counterparts, except when $a^{CV}$ is very large. This is a distinctive feature of greedy strategies as they look for local optima. The clairvoyant vision allows the attacker to allocate its resources more wisely, for example by investing and paying extra perturbation costs for higher future rewards. This is not possible for the greedy attacker, as its vision is limited to the following training step. Substantially, though, the scattered points in Fig. 2-C lie along the diagonal, indicating that greedy attacks are nearly clairvoyant efficient, even for $P = 1$. The match between $a^G$ and $a^{CV}$ corresponds to almost identical greedy and clairvoyant trajectories of the student-teacher and student-target weights overlap, defined respectively as $O_\mu^{st} = w_\mu^{s\,T} w^t / D$ and $O_\mu^{s*} = w_\mu^{s\,T} w^* / D$ (see Fig. 2-D).

Next, we investigate the steady state reached by the student under greedy attacks and the performance of the various attack strategies for three different architectures: the linear regression of Eq. (8), the sigmoidal perceptron $\phi(x; w) = \mathrm{Erf}(z/\sqrt{2})$, with $\mathrm{Erf}(\cdot)$ the error function and $z = w^T x / \sqrt{D}$, and the 2-layer neural network (NN)

$$\phi(x; v, w) = \frac{1}{\sqrt{M}} \sum_{m=1}^{M} v_m \mathrm{Erf}(z_m / \sqrt{2}), \qquad z_m = w_m^T x / \sqrt{D}, \tag{18}$$

with parameters $w_m \in \mathbb{R}^D$ and $v \in \mathbb{R}^M$. We continue to draw all teacher weights from the standard normal distribution, normalizing them to have unitary self-overlap. For the perceptron we set the target weights as $w^* = -w^t$, while for the neural network $w_m^* = w_m^t$ and $v^* = -v^t$. The top row in Fig. 3 shows $\bar{d}_\infty = \lim_{\mu \to \infty} \langle d_\mu \rangle$, i.e. the average steady-state distance reached by the student versus the cost of actions $C$ for greedy attacks. The solid lines show the results obtained from simulations[3]. The yellow shaded area shows $\bar{d}_\infty^G$ form Eq. (13) comprised between values of batch size $P = 1$ (upper contour) and $P = 10$ (lower contour). Note that the three cases show very similar behavior of $\bar{d}_\infty(C, P)$ and are in excellent agreement with the linear TSA prediction.

---

[3]Practically, this involves solving numerically the minimization problem in Eq. (7) on a discrete action grid in the range $[a_{\min}, a_{\max}]$. This is done at each SGD step $\mu$, and it requires simulating the transition $\mu \to \mu + 1$ to estimate the average nefarious cost $\mathbb{E}_x[g_{\mu+1}^{nef}]$ sampling input data from a buffer of past observations.

In order to compare the efficacy of all the attack strategies presented in Sec. 2.4, we consider the *running cost*, defined as $g_\mu = g_\mu^{\text{per}} + g_\mu^{\text{nef}}$. More precisely, we compute the steady-state average $\bar{g}_\infty$ over multiple data streams. The bottom row in Fig. 3 shows this quantity for the corresponding architecture of the top row. The evaluation of the clairvoyant attack is missing for the NN architecture, as this approach becomes unfeasible for complex models due to the high non-convexity of the associated minimization problem. Similarly, reinforcement learning attacks are impractical for large architectures. This is because the observation space of the RL agent, given by the input $x$ and the parameters of the model under attack, becomes too large to handle for neural networks. Therefore, for the NN model we used a TD3 agent that only observes the read-out weights $v$ of the student network; hence the asterisk in RL$^*$. We observe that the constant and clairvoyant strategies have respectively the highest and lowest running costs, as expected. Greedy attacks represent the second-best strategy, with an average running cost just above that of clairvoyant attacks. While we do not explicitly show it, it should also be emphasized that the computational costs of the different strategies vary widely; in particular, while greedy attacks are relatively cheap, RL attacks require longer data streams and larger computational resources, since they ignore the learning dynamics. We refer to Fig. 8, 9 in Appendix D for a comparison between greedy and RL action policies, and for a visual representation of the convergence in performance of the RL algorithm vs number of training episodes.

## 4.2    Experiments on real data

In this section, we explore the behaviour of online data poisoning in an uncontrolled scenario, where input data elements are not i.i.d. and are correlated with the teacher weights. Specifically, we consider two binary image classification tasks, using classes '1' and '7' from MNIST, and 'cats' and 'dogs' from CIFAR10, while attacks are performed using the greedy algorithm. We explore two different setups. First, we experiment on the MNIST task with simple architectures where all parameters are updated during the perturbed training phase. For this purpose, we consider a logistic regression (LogReg) and a simple convolutional neural network, the LeNet of [27]. Then, we address the case of *transfer learning* for more complex image-recognition architectures, where only the last layer is trained during the attacks, which constitutes a typical application of large pre-trained models. Specifically, we consider the visual geometry group (VGG11) [28] and residual networks (ResNet18) [29], with weights trained on the ImageNet-1k classification task. For all architectures, we fix the last linear layer to have 10 input and 1 output features, followed by the Erf activation function.

**Preprocessing**. For LogReg, we reduce the dimensionality of input data to 10 via PCA projection. For LeNet, images are resized from 28x28 to 32x32 pixels, while for VGG11 and ResNet18 images are resized from 32x32 to 224x224 pixels. Features are normalized to have zero mean and unit variance, and small random rotations of maximum 10 degrees are applied to each image, thus providing new samples at every step of training.

**Learning**. The training process follows the TSA paradigm: we train a teacher network on the given classification task, and we then use it to provide (soft) labels to a student network with the same architecture. As before, we consider two learning phases: first, the attacker is silent, and it calibrates $\tilde{\gamma}$. Then attacks start and the student receives perturbed data. For consistency with our teacher-student analysis, we consider the SGD optimizer with MSE loss; see Fig. 10 in Appendix D for a comparison between SGD and Adam-based learning dynamics. To keep the VGG11 and ResNet18 models in a regime where online learning is possible, we use the optimizer with positive weight decay.

The findings are presented in Fig. 4, where panels A and B represent the outcomes of the full training and transfer learning experiments, respectively. Each panel consists of two rows: the top row displays the dependence of the relative student-teacher distance on the cost of action $C$ and batch size $P$. The orange area depicts $\bar{d}_\infty^{\text{G}}$ from Eq. (13) for $P \in [1, 10]$, which we included for a visual comparison. The bottom row shows the corresponding average steady-state accuracy of the student. In general, all learners exhibit comparable behavior to the synthetic teacher-student setup in terms of the relationship between the average steady-state distance $\bar{d}_\infty$ and $C$. Additionally, all experiments demonstrate a catastrophic transition in classification accuracy when the value of $C$ surpasses a critical threshold. The impact of batch size $P$ on the $\bar{d}_\infty$ is also evident across all experiments. The effect of the weight decay used in the transfer learning setting is also visible: the value of $\bar{d}_\infty$ remains larger than zero even when $C$ is very large (and attacks are effectively absent). Similarly, for very low values of $C$, the relative distance remains below 1. We note that calibrating $\tilde{\gamma}$ takes a long time for complex models updating all parameters, so we provide single-run evaluations for LeNet.

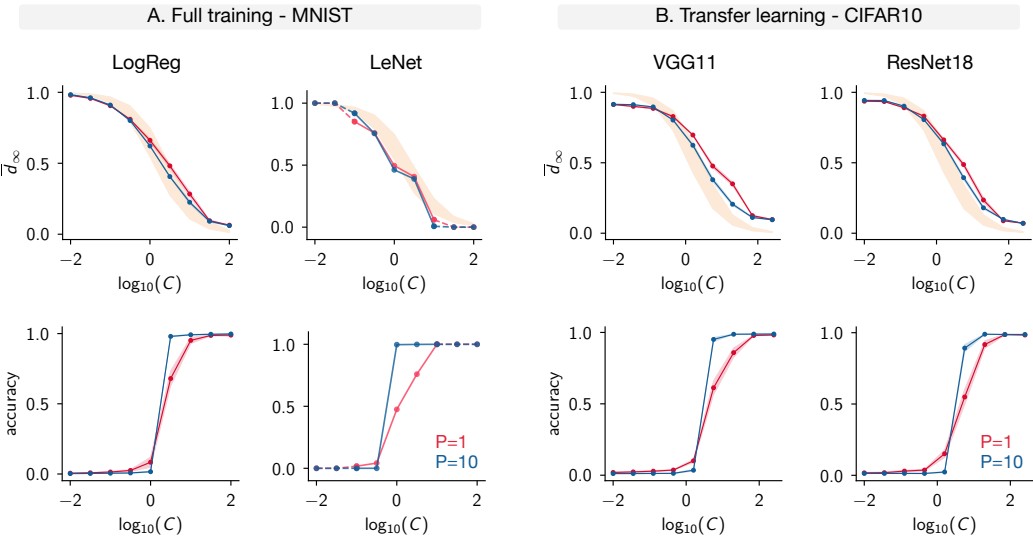

Figure 4: ***Empirical results for the TSA problem with real data***. A, top row: average steady-state distance vs cost of action $C$, for greedy attacks on fully trained logistic regression (LogReg) and LeNet architectures. The orange area represents the range of solutions of $d_\infty^G$ (13) for $P \in [1, 10]$. Bottom row: average steady-state accuracy vs $C$ for the architectures of the top row. B: same quantities as shown in A for transfer learning experiments on VGG11 and ResNet18. Averages were performed over 10 data streams of $10^5$ batches and over the last $10^3$ steps for each data stream. Parameters: $D = 10$, $\eta = 0.02 \times D$ (LogReg, VGG11, ResNet18), $\eta = 0.01$ (LeNet), $a \in [0, 1]$.

## 5   Conclusions

Understanding the robustness of learning algorithms is key to their safe deployment, and the online learning case, which we analyzed here, is of fundamental relevance for any interactive application. In this paper, we explored the case of online attacks that target data labels. Using the teacher-student framework, we derived analytical insights that shed light on the behavior of learning models exposed to such attacks. In particular, we proved that a discontinuous transition in the model's accuracy occurs when the strength of attacks exceeds a critical value, as observed consistently in all of our experiments. Our analysis also revealed that greedy attack strategies can be highly efficient in this context, especially when applying sample-specific perturbations, which are straightforward to implement when data stream in small batches.

A key assumption of our investigation is that the attacker tries to manipulate the long-run behavior of the learner. We note that, for complex architectures, this assumption may represent a computational challenge, as the attacker's action induces a slowdown in the dynamics. Moreover, depending on the context, attacks may be subject to time constraints and therefore involve transient dynamics. While we have not covered such a scenario in our analysis, the optimal control approach that we proposed could be recast in a finite-time horizon setting. We also note that our treatment of online data poisoning did not involve defense mechanisms implemented by the learning model. Designing optimal control attacks that can evade detection of defense strategies represents an important avenue for future research.

Further questions are suggested by our results. While the qualitative features of our theoretical predictions are replicated across all real-data experiments that we performed, we observed differences in robustness between different algorithms. For standard, test-time adversarial attacks, model complexity can aggravate vulnerability [30, 31], and whether this happens in the context of online attacks represents an important open question. Another topic of interest is the interaction of the poisoning dynamics with the structure of the data, and with the complexity of the task. Finally, we remark that our analysis involved sequences of i.i.d. data samples: taking into account temporal correlations and covariate shifts within the data stream constitutes yet another intriguing challenge.

**Compute**. We used a single NVIDIA Quadro RTX 4000 graphics card for all our experiments.

## Acknowledgements

SG and GS acknowledge co-funding from Next Generation EU, in the context of the National Recovery and Resilience Plan, Investment PE1 – Project FAIR "Future Artificial Intelligence Research". This resource was co-financed by the Next Generation EU [DM 1555 del 11.10.22].

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

# A Attack strategies: algorithms

Table 1 summarizes the attack strategies introduced in Sec. 2.4. For clarity, we present the algorithms for data streaming in batches of size $P = 1$. We recall that the action value $a^{\mathrm{c}}$ of constant attacks and the greedy future weight $\tilde{\gamma}$ can be optimized by sampling future data streams from past observations, and using the SGD update rule (5) to simulate the future dynamics. Similarly, the attacker can sample past observations to calibrate the the RL policy $a^{\mathrm{RL}}(x, \theta)$.

---

**Algorithm 1** Online label attacks (batch size $P = 1$)

---

*Constant attacks*
1: **optimize** $a^{\mathrm{c}}$
2: **repeat**
3:     $a_\mu \leftarrow a^{\mathrm{c}}$
4:     $y_\mu^\dagger \leftarrow y_\mu^{\mathrm{t}} + (y_\mu^* - y_\mu^{\mathrm{t}})a_\mu$
5:     $y_\mu^{\mathrm{s}} \leftarrow \phi^{\mathrm{s}}(x_\mu; \theta_\mu^{\mathrm{s}})$
6:     $\theta_{\mu+1}^{\mathrm{s}} \leftarrow \theta_\mu^{\mathrm{s}} - \eta \nabla_{\theta_\mu^{\mathrm{s}}} \mathcal{L}(y_\mu^{\mathrm{s}}, y_\mu^\dagger)$
7: **until** end of stream

*Greedy attacks*
1: **optimize** $\tilde{\gamma}$
2: **repeat**
3:     **observe** clean batch $(x, y^{\mathrm{t}})_\mu$, state $\theta_\mu^{\mathrm{s}}$
4:     $a_\mu \leftarrow \mathrm{argmin}\langle g_\mu^{\mathrm{per}} + \tilde{\gamma} g_{\mu+1}^{\mathrm{nef}}\rangle$
5:     $y_\mu^\dagger \leftarrow y_\mu^{\mathrm{t}} + (y_\mu^* - y_\mu^{\mathrm{t}})a_\mu$
6:     $y_\mu^{\mathrm{s}} \leftarrow \phi^{\mathrm{s}}(x_\mu; \theta_\mu^{\mathrm{s}})$
7:     $\theta_{\mu+1}^{\mathrm{s}} \leftarrow \theta_\mu^{\mathrm{s}} - \eta \nabla_{\theta_\mu^{\mathrm{s}}} \mathcal{L}(y_\mu^{\mathrm{s}}, y_\mu^\dagger)$
8: **until** end of stream

*Reinforcement learning attacks*
1: **calibrate** policy $a^{\mathrm{RL}}(x, \theta)$
2: **repeat**
3:     **observe** clean batch $(x, y^{\mathrm{t}})_\mu$, state $\theta_\mu^{\mathrm{s}}$
4:     $a_\mu \leftarrow a^{\mathrm{RL}}(x_\mu, \theta_\mu^{\mathrm{s}})$
5:     $y_\mu^\dagger \leftarrow y_\mu^{\mathrm{t}} + (y_\mu^* - y_\mu^{\mathrm{t}})a_\mu$
6:     $y_\mu^{\mathrm{s}} \leftarrow \phi^{\mathrm{s}}(x_\mu; \theta_\mu^{\mathrm{s}})$
7:     $\theta_{\mu+1}^{\mathrm{s}} \leftarrow \theta_\mu^{\mathrm{s}} - \eta \nabla_{\theta_\mu^{\mathrm{s}}} \mathcal{L}(y_\mu^{\mathrm{s}}, y_\mu^\dagger)$
8: **until** end of stream

*Clairvoyant attacks*
1: **observe** state $\theta_1^{\mathrm{s}}$, stream $S = \{(x, y^{\mathrm{t}})_\mu\}_{\mu=1}^T$
2: **find** $\{a_\mu\}_{\mu=1}^T = \mathrm{argmin}\, G(S, \theta_1^{\mathrm{s}})$
3: **repeat**
4:     $y_\mu^\dagger \leftarrow y_\mu^{\mathrm{t}} + (y_\mu^* - y_\mu^{\mathrm{t}})a_\mu$
5:     $y_\mu^{\mathrm{s}} \leftarrow \phi^{\mathrm{s}}(x_\mu; \theta_\mu^{\mathrm{s}})$
6:     $\theta_{\mu+1}^{\mathrm{s}} \leftarrow \theta_\mu^{\mathrm{s}} - \eta \nabla_{\theta_\mu^{\mathrm{s}}} \mathcal{L}(y_\mu^{\mathrm{s}}, y_\mu^\dagger)$
7: **until** end of stream

---

# B Linear TSA problem: optimal solution

In the limit of very large batches, batch averages of the nefarious cost $g_\mu^{\mathrm{nef}}$ and loss $\mathcal{L}_\mu$ in Eqs. (2, 5) can be approximated with averages over the input distribution. For a general architecture $\phi$, and indicating with t, s, and $*$ the teacher, student, and attacker's target, respectively, we can write

$$g_\mu^{\mathrm{nef}} = \frac{1}{2}\left(I_\mu(\mathrm{s}, \mathrm{s}) - 2I_\mu(\mathrm{s}, *) + I_\mu(*, *)\right), \tag{19}$$

$$\nabla_{\theta^{\mathrm{s}}} \mathcal{L}_\mu = \frac{1}{2}I'_\mu(\mathrm{s}, \mathrm{s}) - I'_\mu(\mathrm{s}, \mathrm{t})(1 - a_\mu) + I'_\mu(\mathrm{s}, *)a_\mu, \tag{20}$$

where $I(a, b) = \mathbb{E}_x\left[\phi^a(x)\phi^b(x)\right]$ for two models, $a$ and $b$, with the same architecture but different parameters $\theta^a$, $\theta^b$, and $I' = \nabla_{\theta^{\mathrm{s}}} I$. In this case, the optimal control problem (4) is deterministic, and we can write it in continuous time ($\eta \to 0$) as

$$a_\mu^{\mathrm{opt}} = \underset{a_\mu}{\mathrm{argmin}} \int_0^\infty \mathrm{d}\mu \gamma^\mu (g_\mu^{\mathrm{per}} + g_\mu^{\mathrm{nef}}), \tag{21}$$

where $g_\mu^{\mathrm{per}} = \tilde{C}a_\mu^2/2$. Note that $\mu$ now is a continuous time index, and the above minimization problem considers all functions $a_\mu \in [a_{\min}, a_{\max}]$ for $\mu \in [0, \infty)$. In the following, we will assume that the action space is unbounded, i.e. $a_\mu \in (-\infty, \infty)$, and that the elements of $x$ are i.i.d. samples with zero mean and variance $\sigma^2$. In the linear TSA problem, where $\phi(x; w) = w^{\mathrm{T}}x/\sqrt{D}$, the above Eqs. (19, 20) simplify to

$$g_\mu^{\mathrm{nef}} = \frac{\sigma^2}{2D}|\Delta w_\mu^{\mathrm{s}*}|^2, \quad \nabla_{w^{\mathrm{s}}} \mathcal{L}_\mu = \frac{\sigma^2}{D}\left(\Delta w_\mu^{\mathrm{st}} + \Delta w^{\mathrm{t}*}a_\mu\right), \tag{22}$$

where $\Delta w^{ab} = w^a - w^b$, and $|\cdot|^2$ the squared 2-norm. We also find

$$\tilde{C} = \frac{\sigma^2}{D}|\Delta w^{\mathrm{t}*}|^2 C. \tag{23}$$

We shall recall that in the TSA problem each batch is used only once during training. This assumption guarantees that inputs $x_\mu$ and weights $w_\mu^{\mathrm{s}}$ are uncorrelated, a necessary condition to derive the above expressions. We solve this problem with a Lagrangian-based approach: the Lagrangian $L$ is obtained by imposing the constraint $\dot{w}_\mu^{\mathrm{s}} = -\nabla_{w^{\mathrm{s}}}\mathcal{L}_\mu$, at all times, with the co-state variables $\lambda_\mu \in \mathbb{R}^D$:

$$L = \int_0^\infty \mathrm{d}\mu\gamma^\mu(g_\mu^{\mathrm{per}} + g_\mu^{\mathrm{nef}}) - \lambda_\mu^{\mathrm{T}}(\dot{w}_\mu^{\mathrm{s}} + \sigma^2\left(\Delta w_\mu^{\mathrm{st}} + \Delta w^{\mathrm{t}*}a_\mu\right)/D). \tag{24}$$

The Pontryagin's necessary conditions for optimality $\partial_{a_\mu}L = 0$, $[\partial_{w_\mu^{\mathrm{s}}} - \partial_\mu\partial_{\dot{w}_\mu^{\mathrm{s}}}]L = 0$, and $\partial_{\lambda_\mu}L = 0$ give the following set of equations coupling $a_\mu$, $w_\mu^{\mathrm{s}}$, and $\lambda_\mu$:

$$\gamma^\mu\tilde{C}a_\mu - \lambda_\mu^{\mathrm{T}}\sigma^2\Delta\omega^{\mathrm{t}*} = 0,$$
$$\gamma^\mu\sigma^2\Delta\omega_\mu^{\mathrm{s}*}/D + \dot{\lambda}_\mu - \sigma^2\lambda_\mu = 0,$$
$$\dot{w}_\mu^{\mathrm{s}} + \sigma^2(\Delta w_\mu^{\mathrm{st}} + \Delta w^{\mathrm{t}*}a_\mu)/D = 0. \tag{25}$$

The above system of ODEs can be solved by specifying the initial condition $w_0^{\mathrm{s}}$ and using the transversality condition $\lim_{\mu\to\infty}\lambda_\mu = 0$. At steady state, we obtain the following equation for $w_\infty^{\mathrm{s}}$:

$$\Delta w_\infty^{\mathrm{st}} + \sigma^2\Delta w_\infty^{\mathrm{s}*\,\mathrm{T}}\Delta w^{\mathrm{t}*}\Delta w^{\mathrm{t}*}/\tilde{C}D = 0, \tag{26}$$

which admits only one (optimal) solution. We find it as a linear combination of $w^{\mathrm{t}}$ and $w^*$:

$$w_\infty^{\mathrm{s}} = w^{\mathrm{t}} - a_\infty^{\mathrm{opt}}\Delta w^{\mathrm{t}*}, \quad a_\infty^{\mathrm{opt}} = (C+1)^{-1}. \tag{27}$$

From the definition of $d_\mu$ (6), it follows that

$$d_\infty^{\mathrm{opt}} = a_\infty^{\mathrm{opt}}. \tag{28}$$

As a final remark, we observe that the same procedure can in principle be applied to TSA problems with non-linear architectures, as long as an explicit expression for $I(a, b)$ is available. However, one may find multiple steady-state solutions, and obtaining guarantees for optimality is often hard [32].

## B.1 Optimal greedy solution

For the linear TSA problem, and in the limit of large batches, the discrete-time equation (7) for greedy actions simplifies to

$$a_\mu^{\mathrm{G}} = \operatorname*{argmin}_{a_\mu}\left[\frac{1}{2}\tilde{C}a_\mu^2 + \tilde{\gamma}\frac{\sigma^2}{2D}|w_{\mu+1}^{\mathrm{s}}(a_\mu) - w^*|^2\right]. \tag{29}$$

The expression of $w_{\mu+1}^{\mathrm{s}}(a_\mu)$ is given by the update rule, which can be conveniently written as

$$w_{\mu+1}^{\mathrm{s}} = v_1 + v_2 a_\mu, \tag{30}$$

where $v_1 = w_\mu^{\mathrm{s}} - \eta\sigma^2\Delta w_\mu^{\mathrm{st}}/D$, and $v_2 = -\eta\sigma^2\Delta w^{\mathrm{t}*}/D$. The solution of (29) gives the greedy policy function

$$a_\mu^{\mathrm{G}} = \frac{w^{*\,\mathrm{T}}v_2 - v_1^{\mathrm{T}}v_2}{\frac{\tilde{C}D}{\tilde{\gamma}} + |v_2|^2}. \tag{31}$$

The right-hand side of the above equation depends on the future weighting factor $\tilde{\gamma}$. Requiring that the (greedy) update rule $w_{\mu+1}^{\mathrm{s}} = v_1 + v_2 a_\mu^{\mathrm{G}}$ gives the optimal steady-state solution of Eq. (27), and using the explicit expression of $\tilde{C}$ (23), one finds the optimal weight

$$\tilde{\gamma} = D/\sigma^2\eta. \tag{32}$$

## B.2 Accuracy vs cost of action

In Sec. 3.1, we addressed the case of attacks in a classification context, with class labels given by the sign of $\phi^{\mathrm{t}}(x)$, and where the attacker aims to swap the labels, so $\phi^* = -\phi^{\mathrm{t}}$. A pointwise estimate of the accuracy is given by

$$S_\mu(x) = \frac{1}{2}\left(\mathrm{sign}(\phi^{\mathrm{s}}_\mu(x)) + \mathrm{sign}(\phi^{\mathrm{t}}(x))\right), \tag{33}$$

and the system dynamics can be characterized by the accuracy of the student, defined as $A_\mu(C, P) = \mathbb{E}_x[S_\mu(x)]$, with $\mathbb{E}_x$ the average over the input distribution. For the linear TSA problem, we have $w^* = -w^{\mathrm{t}}$, and, consistently with the definition of $d_\mu$ (6),

$$w^{\mathrm{s}}_\mu = w^{\mathrm{t}} + d_\mu(w^* - w^{\mathrm{t}}) = (1 - 2d_\mu)w^{\mathrm{t}}. \tag{34}$$

It follows that $S_\mu(x) = 1$ for $d_\mu \leq 0.5$, and $S_\mu(x) = 0$ otherwise. In the deterministic limit of large batches, the optimal steady-state solution (27) gives

$$S_\infty(x) = \frac{1}{2}\left(\mathrm{sign}\left((C-1)\,w^{\mathrm{t}\,\mathrm{T}}x\right) + \mathrm{sign}\left(w^{\mathrm{t}\,\mathrm{T}}x\right)\right). \tag{35}$$

We then have $S_\infty = 0$ for $C < 1$, and $S_\infty = 0$ otherwise, regardless of the value of $x$, giving

$$A_\infty(C) = 1 - H(C - 1), \tag{36}$$

with $H(\cdot)$ the Heaviside step function.

## C Stochastic control with greedy attacks

When the batch size $P$ is finite, the attacker's optimal control problem is stochastic and has no straightforward solution, even for the simplest case of the linear TSA problem. It is however possible to find the average steady-state solution for greedy attacks, as we show in this Appendix. We derived the greedy policy in B.1 for $P \to \infty$, and for i.i.d. input elements with zero mean and variance $\sigma^2$. For a general $P$, the same expression (31) holds, with $v_1$ and $v_2$ now given by

$$v_1 = w^{\mathrm{s}}_\mu - \eta\frac{1}{DP}\sum_{p=1}^{P}\left(\Delta w^{\mathrm{st}\,\mathrm{T}}_\mu x_{\mu p}\right)x_{\mu p},$$

$$v_2 = -\eta\frac{1}{DP}\sum_{p=1}^{P}\left(\Delta w^{\mathrm{t}*\,\mathrm{T}}x_{\mu p}\right)x_{\mu p}, \tag{37}$$

where $\Delta w^{ab} = w^a - w^b$. The first-order expansion of $a^{\mathrm{G}}_\mu$ in powers of $\eta$ reads

$$a^{\mathrm{G}}_\mu = \frac{1}{\tilde{C}DP}\sum_{p=1}^{P}\Delta w^{\mathrm{s}*\,\mathrm{T}}_\mu x_{\mu p}\Delta w^{\mathrm{t}*\,\mathrm{T}}x_{\mu p} + O(\eta), \tag{38}$$

where we have used $\tilde{\gamma} = D/\sigma^2\eta$, which guarantees that in the limit $P \to \infty$ we recover the optimal solution (27). We can use the above action policy in the SGD update rule (30) and take the average over data streams to get the steady-state equation

$$\Delta\bar{w}^{\mathrm{st}}_{\infty l} = \frac{1}{\tilde{C}DP^2}\left(PT_{1l} + P(P-1)T_{\infty l}\right), \tag{39}$$

where we have imposed $\bar{w}^{\mathrm{s}}_\mu = \bar{w}^{\mathrm{s}}_{\mu+1} = \bar{w}^{\mathrm{s}}_\infty$ for $\mu \to \infty$, and where

$$T_{1l} = \mathbb{E}_x\left[\left(\Delta\bar{w}^{\mathrm{s}*\,\mathrm{T}}_\infty x\Delta w^{\mathrm{t}*\,\mathrm{T}}x\right)\left(\Delta w^{\mathrm{t}*\,\mathrm{T}}xx_l\right)\right], \tag{40}$$

$$T_{\infty l} = \mathbb{E}_{x\neq x'}\left[\left(\Delta\bar{w}^{\mathrm{s}*\,\mathrm{T}}_\infty x\Delta w^{\mathrm{t}*\,\mathrm{T}}x\right)\left(\Delta w^{\mathrm{t}*\,\mathrm{T}}x'x'_l\right)\right]. \tag{41}$$

Note that for $P = 1$ and $P \to \infty$ the right-hand side of (39) only involves $T_{1l}$ and $T_{\infty l}$, respectively (hence the labels). So far, we have assumed that the input elements are i.i.d. samples from $\mathcal{P}_x$ with first two moments $m_1 = 0$, and $m_2 = \sigma^2$. We also recall that $x_\mu$ and $w^{\mathrm{s}}_\mu$ are uncorrelated, as each

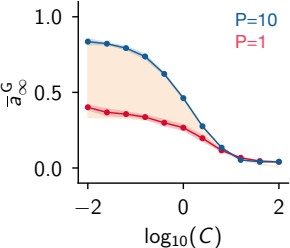

Figure 5: *Average steady-state greedy action*. $\bar{a}^{\mathrm{G}}_{\infty}$ vs cost of actions $C$ for the linear TSA problem with $x \sim \mathcal{N}(0,1)$. The orange area represents the range of solutions (45) for $P \in [1,10]$. The red and blue lines show empirical evaluations for $P=1$ and $P=10$, respectively. Parameters: $D = 10$, $\eta = 10^{-4} \times D$, $a \in [-2,3]$. Averages performed over 10 data streams of $2^4$ batches, and over the last $10^3$ steps in each stream.

batch is used only once in the TSA problem. We now further assume that input elements have third and fourth moments $m_3 = 0$, and $m_4 < \infty$. It is then immediate to verify that $T_{1l} = T_{1l}(m_2, m_4)$ and $T_{\infty l} = T_{\infty l}(m_2)$, showing that finite batch-size effects depend on the kurtosis of the input distribution. Specifically, we find

$$T_{1l} = (m_4 - 3m_2^2)\Delta w_l^{\mathrm{t}*2}\Delta \bar{w}^{*\mathrm{s}}_{\infty l} + m_2^2 |\Delta w^{\mathrm{t}*}|^2 \Delta \bar{w}^{*\mathrm{s}}_{\infty l} + 2m_2^2 \Delta w^{\mathrm{t}*\,\mathrm{T}}\Delta \bar{w}^{*\mathrm{s}}_{\infty}\Delta w_l^{\mathrm{t}*},$$
$$T_{\infty l} = m_2^2 \Delta w^{\mathrm{t}*\,\mathrm{T}}\Delta \bar{w}^{*\mathrm{s}}_{\infty}\Delta w_l^{\mathrm{t}*}. \tag{42}$$

For $\mathcal{P}_x = \mathcal{N}(0,1)$, the first term in $T_{1l}$ disappears, and the above expressions simplify to

$$T_{1l} = |\Delta w^{\mathrm{t}*}|^2 \Delta \bar{w}^{*\mathrm{s}}_{\infty l} + 2\Delta w^{\mathrm{t}*\,\mathrm{T}}\Delta \bar{w}^{*\mathrm{s}}_{\infty}\Delta w_l^{\mathrm{t}*},$$
$$T_{\infty l} = \Delta w^{\mathrm{t}*\,\mathrm{T}}\Delta \bar{w}^{*\mathrm{s}}_{\infty}\Delta w_l^{\mathrm{t}*}. \tag{43}$$

Substituting in Eq. (39) the expressions of $T_{1l}$, $T_{\infty l}$, and $\tilde{C}$ from (23), and looking for a solution of $\bar{w}^{\mathrm{s}}_{\infty}$ as a linear combination of $w^{\mathrm{t}}$ and $w^*$, we find

$$\bar{w}^{\mathrm{s}}_{\infty} = w^{\mathrm{t}} - \bar{d}^{\mathrm{G}}_{\infty}\Delta w^{\mathrm{t}*}, \qquad \bar{d}^{\mathrm{G}}_{\infty} = \left(\left(\frac{P}{P+2}\right)C + 1\right)^{-1}. \tag{44}$$

Note that $\bar{d}^{\mathrm{G}}_{\infty}$ is the relative distance (6) of the student with parameters $\bar{w}^{\mathrm{s}}_{\infty}$. From this result, we can easily find the mean steady-state greedy action by averaging (38) over data streams, giving

$$\bar{a}^{\mathrm{G}}_{\infty} = \frac{f(P)}{f(P)C + 1}, \qquad f(P) = P/(P+2). \tag{45}$$

Fig. 5 shows a comparison of the above result with empirical evaluations, which are in excellent agreement. Together, (44) and (45) demonstrate that greedy attacks become more efficient as the batch size decreases, reducing the distance between the student and the target while using smaller perturbations. We remark that this result is derived using the expression of $\tilde{\gamma}$ from the $P \to \infty$ analysis, as in Eq. (32). In our experiments, this value is near-optimal also for $P$ finite for linear learners, while for non-linear architectures it provides a good first guess during the calibration phase.

## C.1 Sample-specific perturbations

In this section, we address the case where the attacker has finer control over the data labels and can apply sample-specific perturbations, rather than batch-specific ones. The control variable $a$ is $P$-dimensional, with $P$ the size of the streaming batches, and each label $y^{\mathrm{t}}_p$ is perturbed as

$$y^{\dagger}_p = y^{\mathrm{t}}_p(1 - a_p) + y^*_p a_p, \tag{46}$$

while the perturbation cost is $g^{\mathrm{per}}_{\mu} = \tilde{C}|a_{\mu}|^2/2P$. We can write the SGD update rule as

$$w^{\mathrm{s}}_{\mu+1} = \frac{1}{P}\sum_{p=1}^{P} v_{1p} + v_{2p}a_{\mu p}, \tag{47}$$

where now

$$v_{1p} = w^{\mathrm{s}}_{\mu} - \eta\left(\Delta w^{\mathrm{st}\,\mathrm{T}}_{\mu} x_{\mu p}\right) x_{\mu p}/D,$$
$$v_{2p} = -\eta\left(\Delta w^{\mathrm{t}*\,\mathrm{T}}_{\mu} x_{\mu p}\right) x_{\mu p}/D. \tag{48}$$

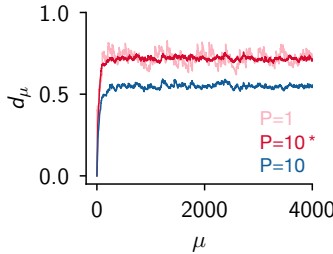

Figure 6: *__Relative distance with sample-specific attacks__*. Student-teacher distance vs SGD step $\mu$ for the linear TSA problem with $x \sim \mathcal{N}(0,1)$. The light-red and blue lines show a single realization of the dynamics for batch-specific control. The dark-red line shows the same quantity for sample-specific, multi-dimensional control. Parameters: $C = 1$, $D = 10$, $\eta = 10^{-2} \times D$, $a \in [-2, 3]$.

We use again the notation $\Delta w^{ab} = w^a - w^b$. Note that (47) reduces to (30) when the control variable is one-dimensional and batch-specific, i.e. $a_{\mu p} = a_\mu \, \forall \, p$. The minimization problem (7) for greedy actions now consists of $P$ coupled equations, one for each control variable $a_{\mu p}$. For i.i.d. input elements with zero mean and variance $\sigma^2$, we find

$$a_{\mu p}^{\mathrm{G}} = \underset{a_{\mu p}}{\operatorname{argmin}} \left[ \frac{1}{2}\tilde{C}|a_\mu|^2 + \tilde{\gamma}\frac{\sigma^2}{2D}|w_{\mu+1}^{\mathrm{s}}(a_\mu) - w^*|^2 \right], \tag{49}$$

and the first-order conditions lead to

$$\tilde{C}a_{\mu p} + \tilde{\gamma}\frac{\sigma^2}{D}\left( \frac{1}{P}\sum_{j=1}^{P} v_{1j}^{\mathrm{T}} + a_{\mu j}v_{2j}^{\mathrm{T}} - w^{*\mathrm{T}} \right) v_{2p} = 0. \tag{50}$$

A straightforward and explicit solution can be found by considering only the first-order terms in $\eta$, effectively decoupling the above system of equations. With $\tilde{\gamma} = D/\sigma^2$, we obtain

$$a_{\mu p}^{\mathrm{G}} \simeq \frac{1}{\tilde{C}D}\Delta w_\mu^{\mathrm{s}*\,\mathrm{T}} x_{\mu p}\Delta w^{\mathrm{t}*\,\mathrm{T}} x_{\mu p}, \tag{51}$$

which coincides with the $P = 1$ solution for batch-specific greedy attacks (38). We can now use this policy in the update rule (47) to investigate the steady state of the system. Once more, we average across data streams with i.i.d. data sampled from $\mathcal{P}_x = \mathcal{N}(0,1)$ and impose that the average student weights $\bar{w}_\mu^{\mathrm{s}}$ reach a steady state for $\mu \to \infty$. This leads to the following $P$-independent solution

$$\bar{w}_\infty^{\mathrm{s}} = w^{\mathrm{t}} - \bar{d}_\infty^{\mathrm{G}}\Delta w^{\mathrm{t}*}, \qquad \bar{d}_\infty^{\mathrm{G}} = (C/3 + 1)^{-1}. \tag{52}$$

Fig. 6 shows empirical evaluations confirming this result. Finally, averaging the policy function (51) across data streams and using the above result for $\bar{w}_\infty^{\mathrm{s}}$ we find

$$\bar{a}_{\infty p}^{\mathrm{G}} \simeq \frac{1/3}{C/3 + 1}. \tag{53}$$

### C.2    Mixing clean and perturbed data

The assumption that the attacker can perturb every data sample in each batch may not always be satisfied. For example, in a federated learning scenario, the central server coordinating the training may have access to a trusted source of clean data that is inaccessible to the attacker. Moreover, the attacker may face a limited computational budget and only be able to process a fraction of samples in the data stream at each timestep. Here, we consider the case where only a fraction $\rho$ of the $P$ samples in each batch is perturbed by the attacker. Moreover, we assume that the attacker ignores the amount of clean samples used for training. Precisely, the attacker considers the update rule

$$\begin{aligned}
\omega_{\mu+1}^{\mathrm{s}} &= \omega_\mu^{\mathrm{s}} - \eta\nabla_{\omega_\mu^{\mathrm{s}}}\mathcal{L}_\mu\left(B_\mu^{\rho P\dagger}\right) \\
&= \omega_\mu^{\mathrm{s}} - \frac{\eta}{D\rho P}\left( \sum_{p=1}^{\rho P}\left(\Delta w_\mu^{\mathrm{st}\,\mathrm{T}} x_{\mu p}\right)x_{\mu p} + a_\mu\sum_{p=1}^{\rho P}\left(\Delta w_\mu^{\mathrm{t}*\,\mathrm{T}} x_{\mu p}\right)x_{\mu p} \right),
\end{aligned} \tag{54}$$

with $\rho P$ perturbed samples, while training follows

$$\begin{aligned}
\omega_{\mu+1}^{\mathrm{s}} &= \omega_\mu^{\mathrm{s}} - \eta\rho\nabla_{\omega_\mu^{\mathrm{s}}}\mathcal{L}_\mu\left(B_\mu^{\rho P\dagger}\right) - \eta(1-\rho)\nabla_{\omega_\mu^{\mathrm{s}}}\mathcal{L}_\mu\left(B_\mu^{(1-\rho)P}\right) \\
&= \omega_\mu^{\mathrm{s}} - \frac{\eta}{DP}\left( \sum_{p=1}^{P}\left(\Delta w_\mu^{\mathrm{st}\,\mathrm{T}} x_{\mu p}\right)x_{\mu p} + a_\mu\sum_{p=1}^{\rho P}\left(\Delta w_\mu^{\mathrm{t}*\,\mathrm{T}} x_{\mu p}\right)x_{\mu p} \right),
\end{aligned} \tag{55}$$

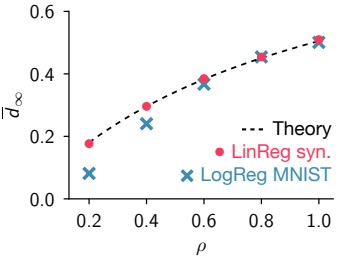

Figure 7: ***Mixing clean and poisoned samples***. Average steady-state distance vs fraction $\rho$ of poisoned samples, under greedy attacks. The dashed line shows the estimate of Eq. (58), and the red dots display the associated empirical evaluations. Blue crosses show empirical results obtained for a logistic regression training on MNIST digits. Parameters: $C = 1$, $P = 100$, $D = 10$, $\eta = 0.1$, $a \in [-2, 3]$. Averages performed over 10 data streams of $10^4$ batches, and over the last $10^3$ steps in each stream.

thus involving $\rho P$ perturbed samples and $(1 - \rho)P$ clean samples. We recall that $\Delta w^{ab} = w^a - w^b$, and that the stream has input data drawn i.i.d. from $\mathcal{P}_x$. Following (54), the greedy policy (38) in this case reads

$$a_\mu^{\mathrm{G}} = \frac{1}{\tilde{C}D\rho P} \sum_{p=1}^{\rho P} \Delta w_\mu^{\mathrm{s*\,T}} x_{\mu p} \Delta w^{\mathrm{t*\,T}} x_{\mu p} + O(\eta). \tag{56}$$

We can now substitute the above expression into (55) to investigate the steady state of the system. As before, we can obtain a steady-state condition for $\bar{w}_\infty^{\mathrm{s}}$ by averaging across data streams and imposing $\bar{w}_\mu^{\mathrm{s}} = \bar{w}_{\mu+1}^{\mathrm{s}} = \bar{w}_\infty^{\mathrm{s}}$ for $\mu \to \infty$. We find

$$\Delta \bar{w}_{\infty l}^{\mathrm{st}} = \frac{1}{\tilde{C}D\rho P^2} \Big( \rho P T_{1l} + \rho P (\rho P - 1) T_{\infty l} \Big), \tag{57}$$

with $T_{1l}$ and $T_{\infty l}$ as in (43) for $\mathcal{P}_x = \mathcal{N}(0, 1)$. It is then immediately verified that

$$\bar{w}_\infty^{\mathrm{s}} = w^{\mathrm{t}} - \bar{d}_\infty^{\mathrm{G}} \Delta w^{\mathrm{t*}}, \qquad \bar{d}_\infty^{\mathrm{G}} = \left( \left( \frac{P}{\rho P + 2} \right) C + 1 \right)^{-1}. \tag{58}$$

Note that for large batches ($\rho P \gg 2$) the above result reduces to $\bar{d}_\infty^{\mathrm{G}} \approx (C/\rho + 1)^{-1}$, and it has a simple and intuitive interpretation: when perturbations contaminate a fraction $\rho$ of the batch samples only, the cost of actions effectively increases by a factor $\rho^{-1}$. Fig. 7 shows the average steady-state distance reached by the student model as a function of $\rho$, demonstrating that empirical evaluations using synthetic data perfectly match the theoretical estimate of Eq. (58). We also performed real data experiments using MNIST images (as in Sec. 4.2), which again exhibit a good qualitative agreement with our theoretical result.

## D   Supplementary figures

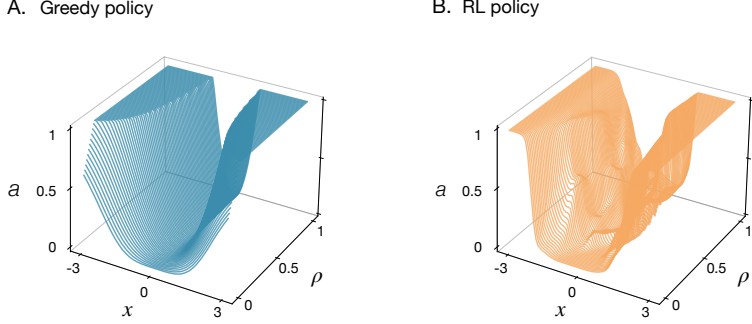

Figure 8: ***Greedy and RL policies***. A: greedy action policy given by Eq. (31) with $v_1$ and $v_2$ as in (37) with $P = 1$, for the linear TSA problem with one-dimensional input. The $x$-axis shows values of the input, while $\rho$ represents the student state expressed as $\omega^{\mathrm{s}} = w^*(1 - \rho) + w^{\mathrm{t}}\rho$. B: best TD3 policy function found for this problem. Parameters: $C = 1$, $\eta = 0.02$, $a \in [0, 1]$, $w^{\mathrm{t}} = 1$, $w^* = -1$.

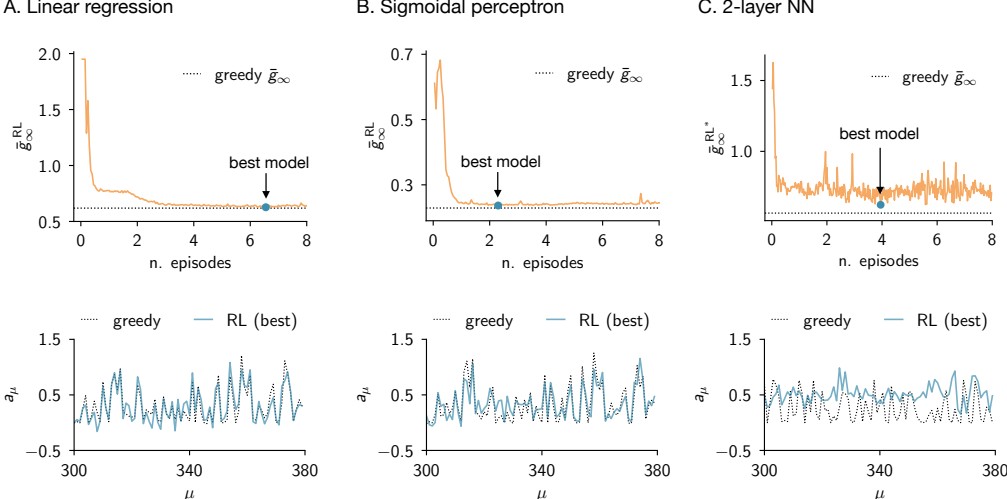

Figure 9: ***Convergence of RL agents***. A, top panel: average steady-state running cost of the TD3 agent vs training episodes, for attacks on the linear regression model. Training episodes have length of $4 \times 10^4$ SGD steps, and policy updates are performed every 100 steps. The dotted line shows the average steady-state running cost of the greedy algorithm. The blue dot indicates the best performance achieved by the TD3 agent. Bottom panel: example of actions performed by the greedy attacker (dotted line) and by the best TD3 agent (blue line) on the same data stream. B, C: same quantities as shown in A, for attacks on the sigmoidal perceptron and 2-layer NN. For the latter case, the TD3 agent observed the read-out weight layer only. The best TD3 agents were employed for the comparison shown in Fig. 3. Parameters: $P = 1$, $C = 1$, $D = 10$, $\eta = 0.02 \times D$, $a \in [-2, 3]$.

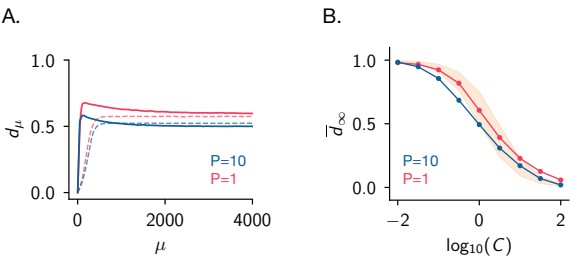

Figure 10: ***Comparison of Adam and SGD dynamics***. A: average relative distance vs training step $\mu$ for a logistic regression model classifying MNIST digits 1 and 7, under label-inversion greedy attacks, for $C = 1$. Solid and dashed lines correspond to training via Adam and SGD, respectively. Average performed over 100 data streams. B: average steady-state relative distance as a function of the cost of actions $C$ and batch size $P$, obtained for the experiment described in panel A, and using the Adam optimizer. The orange shaded area shows the SGD-based theoretical estimate (Eq. (13)) for $P$ in the range $[1, 10]$. Averages performed over 10 data streams of $10^4$ batches, and over the last 1000 steps in each stream. Parameters: $D = 10$, $\eta = 0.1$ (SGD), $\eta = 0.01$ (Adam), $a \in [0, 1]$.

