# OpenReview forum: "Attacks on Online Learners: a Teacher-Student Analysis"
_NeurIPS.cc/2023/Conference — NeurIPS 2023 poster_

### Official Review · Reviewer_MYtD · 2023-07-06

**Soundness:** 3 good
**Presentation:** 2 fair
**Contribution:** 2 fair
**Rating:** 5
**Confidence:** 3

**Summary:**

This paper sheds light on the vulnerabilities of online learners to adversarial attacks and provides insights into the manipulation of learning dynamics. The findings highlight the need for robust defenses against such attacks and contribute to the growing body of research on data poisoning attacks in machine learning.


**Strengths:**

1. The paper provides a theoretical analysis of online data poisoning in the teacher-student setup, which is a popular framework for studying machine learning models in a controlled way.
2. The paper compares different attack strategies and finds that properly calibrated greedy attacks can be as effective as attacks with full knowledge of the incoming data stream.
3. The authors empirically study online data poisoning on real datasets, such as MNIST and CIFAR10, using various architectures like LeNet, ResNet, and VGG.


**Weaknesses:**

1. This paper primarily focuses on linear regression models and does not extensively explore the impact of adversarial attacks on other types of machine learning models.
2. This paper does not provide specific examples of real-world scenarios where adversarial attacks on online learners can have significant consequences.
3. This paper lacks extensively discuss potential defense mechanisms or countermeasures to mitigate the impact of these attacks.
4. This paper does not thoroughly compare its findings with existing literature on attacks on online learners.
5. The practicality studied threat model should be discussed with more details.


**Questions:**

1. How do different attack strategies compare in terms of efficacy?
2. Are attacks subject to time constraints and do they involve transient dynamics?
3. Does the analysis take into account temporal correlations within the data stream?
4. The model's robustness improves or decreases of over-parametrization?


**Limitations:**

Missing limitation and negative societal impact discussion in this paper.

---

> ### Author Rebuttal · Authors · 2023-08-09
>
> We thank the Reviewer for the constructive criticism and the relevant questions. Please see below for our response.
>
> **Weaknesses**
>
> This paper primarily focuses on linear regression models and does not extensively explore the impact of adversarial attacks on other types of machine learning models.
>
> * The problem of online adversarial attacks has not been addressed theoretically before, so decided to start with the simplest possible setting: linear models. This allowed us to obtain exact and explicit results, which can help develop intuitions about the problem. Note that various deep learning phenomena have been investigated by means of linear models recently (feed-forward neural networks [1], self-supervised learning [2,3], pre-training [4], implicit bias[5-6]).
> We agree with the reviewer that it is crucial to verify if the insights obtained theoretically carry over to more realistic settings. For this reason, we extensively performed experiments with deep networks (LeNet, VGG, ResNet) on MNIST and CIFAR10. We found that key insights from our theoretical analysis, e.g. the steady-state behavior of the learner under attack, are replicated in these settings.
> Additionally, we have now expanded our empirical evaluations using Adam (see Fig. A, B in the attached document), thus exploring even further more realistic use-cases.
>
> Examples of real-world scenarios.
>
> * Several practical use cases motivate our analysis. In federated learning, for example, malicious nodes in the federation can modify the labels before sending the model updates to the central server for aggregation. Label attacks can also occur in online learning settings where models are fine-tuned on the fly using user-generated data, such as email spam filters, product recommendation engines, chatbots, and fake review detectors. The online setting makes adversarial attacks extremely dangerous, since the attackers can monitor the effect of previous poisoning on the learner and make adaptive decisions on how to poison next.
>
> Defense mechanisms.
>
> * Our contribution aims to establish the fundamental behavior of online learners under attack. This problem has not been addressed before theoretically, and the analysed baseline already exhibits rich and non-trivial phenomena. Therefore, a comprehensive analysis of defence strategies would extend beyond the scope of the current paper. Nevertheless, we note that our analysis of attacks that only perturb a fraction $\rho$ of the data (see answer to reviewer 7DCo) is suitable for describing a defence strategy combining the poisoned data with a trusted source of clean data. We found that the attack strength decreases by a factor $\rho$ in this case.
>
> Compare findings with existing literature.
>
> * We did an extensive search for studies addressing the problem of online label poisoning but could not find any relevant paper for benchmarking our methods. We will be happy to consider any additional reference in case the Reviewer had specific suggestions.
>
> The practicality studied threat model should be discussed with more details.
>
> * We consider targeted attacks that aim for the ML model to learn a “nefarious” target function $\phi^*$. The attacker perturbs the data labels to do so, so our setting can be considered a generalisation of label-flipping attacks. We consider attackers with different adversarial capabilities. RL agents ignore the learning algorithm used for training the ML model and see the model architecture. Greedy attacks assume full knowledge of the learning rule and model parameters.
>
> **Questions**
>
> How do different attack strategies compare in terms of efficacy?
>
> * We compared the efficacy of different attack strategies in terms of the average running costs achieved at steady state (see Eq. (16) in the paper). The results are shown in Fig. 3 (paper, main text). Our evaluations indicate that clairvoyant (CV) attacks are the most efficient. This is expected as CV attacks have the advantage of seeing the future data stream upfront. Remarkably, greedy attacks almost match the clairvoyant efficiency in all settings that we considered. RL attacks perform well too when the agent can observe the entire state of the system, and become less effective otherwise (Fig. C, attached document).
>
>
> Transient dynamics and temporal correlations.
>
> * These are indeed interesting scenarios to be explored. They however need a significantly different setup compared to our analysis, where a key assumption is that the attacker tries to minimise the expected cost at steady state. Nevertheless, we shall note that our optimal control formulation can be re-cast in a finite-horizon scenario, which would be suitable to study time-constrained attacks and data exhibiting correlations in time. We have a comment in the concluding section of the paper that include the above considerations.
>
> Robustness improves or decreases with over-parametrization?
>
> * Our real-data experiments suggest that the robustness of ML models may decrease for highly parametrised architectures compared to simple ones (see the results obtained for a logistic regression and LeNet on the same dataset, Fig. 4-A in the paper). A thorough analysis of the role of over-parametrization represents yet another interesting research direction that could be addressed in a dedicated study.
>
> **Limitations**
>
> * In the concluding section, we present several limitations of our analysis, including the assumption of stationary data, the infinite-horizon setup, and the unexplored role of model complexity. In the final version of the paper, we will expand this section to include the limitations associated with the type of attacks (targeting the labels), and the threat model (white/black box attacks).
>
> References
>
> [1] Saxe et al. ICLR ‘13;
>
> [2] Tian, Chen, Ganguli, ICML ‘21;
>
> [3] Jing, Vincent, LeCun, Tian, ICLR ‘22;
>
> [4]  Wu, Zou, Braverman, Gu, Kakade, NeurIPS ‘22;
>
> [5] Moroshko, Woodworth, Gunasekar, Lee, Srebro, Soudry, NeurIPS ‘21;
>
> [6] Li et al. NeurIPS ‘22.

---

### Official Review · Reviewer_7DCo · 2023-07-06

**Soundness:** 3 good
**Presentation:** 2 fair
**Contribution:** 3 good
**Rating:** 5
**Confidence:** 4

**Summary:**

Data poisoning attacks have been extensively studied in the offline setting, while poisoning attacks in the online setting have received little attention. In the online setting, the attacker is forced to craft attacks exploiting the data stream, taking into account the state of the model and the possible future data stream to divert the model. This work aims to study theoretically and empirically the learning dynamics of an online learner subject to a poisoning attack in the white-box setting, i.e., the attacker knows the architecture and the current parameters of the target model. The problem of finding the best attack policy is formalized as a stochastic optimal control problem and is theoretically studied in the teacher-student setup. From that, the authors were able to obtain results about the steady state of an attacked linear regression model, i.e., the model weights, the optimal action and the distance of the target model depending on the cost of the attack, supposing both infinite and finite size batches. Furthermore, the work shows analytically that a greedy strategy can perform quite optimally. The experimental evaluation confirms the theoretical results on synthetic data and explores the dynamics of online data poisoning attacks on real data and non-linear predictors, showing that the theoretical findings almost hold in more complex scenarios.


**Strengths:**

This paper makes a valuable contribution to the literature on data poisoning attacks, where there has been limited work in online settings. In particular:

- The idea of analyzing the attacker’s control problem from the perspective of the teacher-student framework represents the strong main novelty of the paper, since the formulation of the attack problem as an optimal control problem has been discussed before. The analysis allows the authors to derive interesting results about the dynamics of the learning process of the model and the steady state of a linear regressor that have not previously been discussed in the literature. It is interesting to see that above a certain attack strength the accuracy of the attacked model may drop dramatically and perturbations above the threshold may not be sufficient to realize the attack.

- The set of considered attack strategies sufficiently covers the possible options available to the attacker. The idea of showing that a greedy attack can be as effective as an attack with full knowledge is valuable.

- The theoretical results are confirmed empirically and punctually. Moreover, the experimental results on non-linear architectures and real data suggest that the theoretical results may also hold for more complex scenarios, that’s an interesting finding.

- The derivation of the theoretical results is explained in the appendix and the well-organized code to reproduce the experimental results is available (even though more instructions to reproduce the results may be added, like the Python version used by the authors to run the experiments).


**Weaknesses:**

- The theoretical analysis is limited to consider linear regression as the student model, the standard normal distribution as input distribution and an infinite time horizon, so the theoretical results are a bit limited. However, it is recognized by the authors that more general results are missing because deriving them in a more general scenario is hard. Furthermore, the authors tried to derive some intuitions about the results in a more complex scenario through the experimental evaluation. Covering the theoretical analysis in a more complex setting and with a finite time horizon is an interesting future work to explore.

- It is not clearly motivated why the attacker can only modify the labels instead of the entire inputs and why the attacker should be able to apply the same perturbation to the entire batch (which turns out to be a very strong assumption). Data poisoning attackers should assume little control over the data to be considered practical [P1, P2].

- The presentation of the theoretical results is a bit confusing since it is mixed with the presentation of part of the theoretical results.

- The presentation of the attack strategies and their implementation lack details that cannot be found in the appendix. See the “Questions” section for more details.

- From the paper, it is not evident what the take-home message is and what suggestions are provided to improve robustness in stream pipelines or make attackers more stealthy. The paper presents numerous useful analyses and extensive experimental settings but lacks a "Take-home message/Discussion" section that summarizes the overall findings and suggestions. While the authors have reported some of these in the conclusion section, I believe they can be further expanded with higher priority/attention.

[P1] Back to the Drawing Board: A Critical Evaluation of Poisoning Attacks on Federated Learning. In IEEE Symposium on Security and Privacy (2022).

[P2] Wild Patterns Reloaded: A Survey of Machine Learning Security against Training Data Poisoning. ACM Computing Surveys (2022).



**Questions:**

Major questions:

- Why are the clairvoyant and RL attacks infeasible to deploy for complex models? Is it due to the time required for the calibration? This is not very clear from the text.

- In Eq. 1, please clarify why it should be important to model an attacker that “under/overshoots”. Why can the attacker obtain a perturbed label that resides outside the interval between the clean and target labels?

- I think that more details about the RL attack are needed. How is the problem of finding the optimal sequence of actions mapped to the RL problem? Which are the reasons for choosing to utilize TD3 (and a deep learning policy in general) instead of other RL approaches?

- Which is the utility of gamma^tilde in the objective function minimized by the greedy attack? Is it used to address the fact that the attacker should not try the most effective attack at the subsequent time step, but she has to take into account all the future streams of instances? How does the calibration of gamma^tilde work? I invite the authors to add more details about this.

- I suggest that the authors add a "Take-home-messages/Discussion” section that summarizes the theoretical and empirical results and highlights how the results can be useful for designing a poisoning attack or defending against it.

- The authors should clarify their threat model in the paper and where it may apply.


Minor questions:

 - Are the results of Figures 1-B and 1-C obtained from the experimental evaluation? If yes, the setting used for obtaining these results should be indicated in the text. I may have missed it, so I kindly ask the authors to point me out the lines containing this information if it is included in the paper.

- Should g^nef be at the place of one of the g^pre in eq. 16?

Eventually, I kindly ask the authors to fix the links to the equations that appear in the text, since they don’t work at the moment.


**Limitations:**

The authors have briefly discussed some limitations of their work in the conclusion section and they have clearly stated that extending their analysis to non-linear architectures and different probability distributions of the inputs is challenging. However, I think that a discussion about the limitations of the threat model considered is missing and I invite the authors to cover this point, by discussing, for example, why the attacker should only modify the labels of the instances. Moreover, I ask the authors to include a “take-on-messages/discussion” section to summarize and clarify the obtained results and their implications.

---

> ### Author Rebuttal · Authors · 2023-08-09
>
> We thank the Reviewer for the constructive criticism and the relevant questions. Please see below for our response.
>
> **Weaknesses**
>
> Attacker only modifies the labels instead of the entire inputs.
>
> * The case of online poisoning of the labels has not been addressed before, despite its relevance in several practical applications (federated learning, learning from user-generated data, etc), so our aim is to provide a first contribution to fill this gap. Note that attacks on the labels have a reduced computational cost compared to high-dimensional adversarial perturbations of the inputs, which makes label poisoning more convenient in a streaming scenario.
>
> Perturbation of the entire batch.
>
> * While the online setting requires the attacker to keep perturbing the data over time (as otherwise the perturbation may be forgotten by the learner), it is true that the attacker may be able to perturb only a fraction of samples in the batch. Our analytical derivation can be generalised to accommodate this scenario. In the linear TSA problem, we find that greedy attacks lead to an average steady-state distance given by ${(C P / (\rho P + 2) + 1)}^{-1}$, with $\rho$ the fraction of corrupted samples, which for $P$ large reduces to $(C/\rho + 1)^-1$. This result indicates that the attack strength effectively decreases by a factor $\rho$. Our experimental evaluations shown in Fig. E (attached file) support this finding. This result suggests that a simple defence mechanism consisting in mixing poisoned data with clean examples from a trusted source (in case this is available) could improve the model robustness.
>
> Presentation is a bit confusing.
>
> * We will highlight more our theoretical results in a separate, dedicated section in the final version of the paper.
>
> Implementation lacks details that cannot be found in the appendix.
>
> * We thank the reviewer for pointing this out. We will add an Appendix in the final version of the paper to include more details about our implementation and our experiments.
>
> **Questions**
>
> Why Clairvoyant and RL attacks are infeasible?
>
> * The clairvoyant setting is not doable due to the high nonconvexity of the problem. For complex architectures, finding a solution to the equations takes too long to be practical. In the RL approach, the agent observes the state of the system, given by the input data and the learner’s weights, and produces an action according to its policy function. For big architectures, the state space is extremely large, and tuning the RL policy function becomes prohibitively expensive.
>
> * Following the Reviewer’s question, we considered using a TD3 agent that only observes the last-layer weights of the network, significantly reducing the dimension of the state space. The results are shown in Fig. C,D (attached document). The agent performs better than constant attacks, though it does not match the greedy attacks. We will add these evaluations to the final version of the paper.
>
> Clarify why it should be important to model an attacker that “under/overshoots”.
>
> * There is in principle no reason why $a$ should be constrained within a bounded region, as the attacker’s optimization problem includes the perturbation cost proportional to $a^2$. It is up to the attacker, then, to find the best possible sequence of actions, accounting both for the perturbation and the nefarious costs.
>
> More details about the RL attack are needed.
>
> * RL calibrates a policy function to maximise the expected total reward of the agent by interacting with the environment. In our case, the environment is characterised by the current learner’s weights and input data, with rewards given by the negative value of the running cost. A policy that maximises the expected rewards also minimises the running cost of the attacks, finding a solution to the attacker’s optimal control problem. We employed TD3 because it is a powerful agent that requires little tuning of the hyperparameters and can handle continuous action spaces.
>
> More details about $\tilde{\gamma}$.
>
> * Calibrating $\tilde{\gamma}$ helps the greedy attacker overcoming the limitation of a strategy that only looks one step ahead, accounting for long-run effects of the actions. To calibrate $\tilde{\gamma}$, the attacker uses past observations to simulate data streams, and it evaluates the total cost of the simulated trajectory associated with different values of $\tilde{\gamma}$. Averaging over many data streams, the attacker finds the value of $\tilde{\gamma}$ that minimises the (simulated) total expected cost.
>
> The authors should clarify their threat model.
>
> * We consider targeted attacks that aim for the ML model to learn a “nefarious” target function $\phi^*$. The attacker perturbs the data labels to do so, so our setting can be considered a generalisation of label-flipping attacks. We consider attackers with different adversarial capabilities. In particular, RL agents ignore the learning rule and see part or all of the learner's architecture. Greedy attacks assume full knowledge of the learning rule and model parameters.
>
> Missing details of results of Figures 1-B and 1-C.
>
> * For those experiments, with considered streams of $10^4$ batches of size $P=1$, learning rate $\eta=0.1$, and actions $a \in [0, 1]$. We will add these details to the paper.
>
> g^nef at the place of one of the g^pre in eq. 16?
>
> * Yes. Typo fixed.
>
> **Limitations**
>
> We thank the Reviewer for the insightful suggestions. We will expand the concluding section to include the limitations associated with the type of attacks (targeting the labels), and the threat model (white/black box attacks). We will as well include a take-home-message section summarising our results and clarifying their implications, incorporating Reviewers' feedback and including any valuable insight arising from the discussion period.
>
> We hope that our answers have clarified the reviewer’s concerns and will provide grounds for a more favorable assessment of the paper.

---

> > ### Comment · Reviewer_7DCo · 2023-08-21
> >
> > Thanks to the authors for their clear response to my concerns.
> >
> > Similarly to other reviewers, I initially held reservations regarding the considered threat model and the current scope of the proposed work. However, despite these reservations, I have chosen not to lower my original score. This decision is primarily influenced by the significant and noteworthy experimental findings that the authors have presented in their rebuttal.
> >
> > In conclusion, the paper will require several clarifications in its final iteration. These clarifications should encompass aspects such as the delineation of the threat model, underlying assumptions and limitations, and the rationale behind the chosen attack strategy. As pointed out by all reviewers, these elements are pivotal to enhancing the clarity and rigor of the paper. Furthermore, a more seamless integration of the outcomes presented in the main body of the paper with those provided in the appendix is advised for achieving a coherent narrative.

---

> > > ### Author Response · Authors · 2023-08-21
> > >
> > > We thank the reviewer for the response and the positive evaluation of our supplementary findings in the rebuttal. We are pleased that our response effectively addressed the reviewer's concerns. Overall, we believe that our submission has significantly benefited from the reviewers' feedback. In the final version of the manuscript, we will incorporate the clarifications mentioned by the reviewer, as we discussed in our rebuttal.

---

### Official Review · Reviewer_61gD · 2023-07-10

**Soundness:** 2 fair
**Presentation:** 3 good
**Contribution:** 2 fair
**Rating:** 3
**Confidence:** 3

**Summary:**

This paper analyzes the robustness of online learners when the labels of the received data are manipulated. In particular, it analyzes a student-teacher online learning problem where the attacker poisons the labels provided by the teacher before feeding the student the labeled batch. The setup is analyzed both theoretically and experimentally where with the increase of the attack budget, the performance of the learner degrades significantly.

**Strengths:**

The main strengths of this work are:

(1) The paper is well-written and the figures are polished.

(2) The setup is clearly explained and the theoretical analysis on synthetic data support the results.


**Weaknesses:**

There are few weaknesses that I hope to be addressed before getting this paper accepted:

(1) The paper is missing a strong practical motivational example from real world scenarios when attackers have access to manipulate the *labels* before feeding the labeled batches to the learner.

(2) While the proposed problem setup studies an online learning scheme, the attacker is allowed for an unbounded computational budget to manipulate the labels. However, in online learning, the environment can reveal new batches *irrespective* of how efficient the attacker is in manipulating the previously revealed batch. Note that this assumption (given unlimited computation to the attacker) is not realistic and it questions that effectiveness of the proposed attack.

(3) Experimental evaluation are conducted on very small scale problems and datasets. I would expect evaluating the results on online learning datasets such as CLOC [A] and Firehose [B].

(4) The analyzed attackers should be evaluated when a defense mechanism is presented. For instance, would a learner upon convergence still update all model parameters on each received data? What happens if the learner stored labeled examples from the stream before the poisoning phase starts and update on mixed batches?

[A] Online Continual Learning with Natural Distribution Shifts: An Empirical Study with Visual Data, ICCV 2021.

[B] Drinking from a Firehose: Continual Learning with Web-scale Natural Language, 2020.

**Questions:**

In addition to the points raised in the weaknesses part, I have the following question:\

- To guarantee that the attack is imperceptible (in case of perturbing the input), the attacker has to produce the perturbations under budget constraints (e.g. $\|\delta\|_\infty \leq \frac{8}{255}$). How is this constraint translated to the proposed setup?

- The perturbed label in Equation (1) could produce a real number for integer valued labels. Couldn't this be a simple check for the learner to reject such samples from training on them?

**Limitations:**

The authors have not addressed the limitations of this work such as the practicality of the proposed setup, the limited experimental setup (problem and dataset scale) along with how expensive is it to conduct such attacks.

---

> ### Author Rebuttal · Authors · 2023-08-09
>
> We thank the Reviewer for the constructive criticism and the relevant questions. Please see below for our response.
>
> **Weaknesses**
>
> Examples of real world scenarios where attackers have access to manipulate the labels before feeding the labeled batches to the learner.
>
> * **There are several practical use cases where attackers have access to labels before the learner, motivating our analysis**. (i) In federated learning, where data are collected independently by each node in the federation. Malicious nodes could modify the labels before providing the model updates to the central server for aggregation. Even if data are distributed from the central server to the nodes, when collecting the model updates, the central server would have no access to the labels used locally to generate the updates. (ii) Label attacks can also occur in online learning settings where models are fine-tuned on the fly using user-generated data, such as email spam filters, product recommendation engines, chatbots, and fake review detectors. We will add these examples to the introductory section of the manuscript. Note that, in streaming scenarios, the attacker can monitor the effect of previous poisoning on the learner and make adaptive decisions on how to poison next. This adaptability, intrinsic to the online setting, makes attacks on online learners potentially extremely dangerous.
>
>
> ...the attacker is allowed for an unbounded computational budget to manipulate the labels.
>
> * We agree with the reviewer that an unbounded computational budget for attacking the labels is unrealistic. Our proposed attack strategies have a small computational cost for deployment, however, as they only require one forward-and-backward pass on the learner architecture for the greedy attacks, and one forward pass on the TD3 agent for RL-based attacks. Moreover, we are considering attacks that target data labels, which have a reduced computational cost compared to high-dimensional adversarial perturbations of the inputs. These attacks might still be infeasible when data arrives with high frequency, so we will add a comment in the paper to reflect this limitation. To address this problem, we have extended our theoretical analysis to the case where the attacker poisons only a fraction $\rho$ of each data batch (see answer to reviewer 7DCo). The result indicates that the attack strength effectively decreases by a factor $\rho$. Our experiments support this finding (see Fig. E in the attached file).
>
> ... evaluating the results on online learning datasets such as CLOC [A] and Firehose [B].
>
> * We agree with the reviewer that it would be interesting to explore attacks when clean data exhibits a natural covariate shift, as is the case in the CLOC and Firehose data sets, and we will explore this direction in future work. For this paper, our focus is on stationary distributions, and our theoretical analysis is based on this assumption. The goal of our experiments with deep architectures (LeNet, VGG, ResNet) on realistic datasets like MNIST and CIFAR10 was simply to verify our theoretical predictions in cases where data has non-trivial correlations and the optimisation problem is non-convex, which is standard practice in papers with a theory focus at venues like NeurIPS.
>
> What happens if the learner stored labeled examples from the stream before the poisoning phase starts and update on mixed batches?
>
> * We agree that it is important to design defence mechanisms against online adversarial attacks. However, the purpose of this paper was to establish the baseline behaviour of online learners when such attacks are present. This, to the best of our knowledge, was not known and it does already exhibit rich and interesting behaviours. Therefore, a thorough analysis of defence strategies would go beyond the scope of the present paper.
> A simple strategy that mixes (new) poisoned data with clean samples makes the problem equivalent to the case where only a fraction $\rho$ of the data is perturbed by the attacker. Thus, our result mentioned above applies: the attack strength decreases by a factor $\rho$. Note that, to implement this defence mechanism, simply storing data from before the attacks might be challenging - how does the victim know when the attacks start? It would instead be doable in the context of federated learning.
>
> How is this constraint translated to the proposed setup?
>
> * The budget constraint is intrinsic in the setup of the optimal control problem as the attack strength (1/C), which effectively sets the amplitude of the attacks by affecting the perturbation cost. A large attack strength implies large perturbations. The question then is how small C must be to achieve a certain goal (for example making the accuracy low). Similarly, given a value of C, how close do different attack strategies get to a desired target? Our analysis addresses these questions.
>
> ...a real number for integer valued labels. Couldn't this be a simple check for the learner to reject such samples from training on them?
>
> * A simple way to address this limitation is to discretise the actions. For example, greedy attacks could use the action $a^G_{\mu}$ on average, replacing a fraction $\rho=a^{\mathrm{G}}_{\mu}$ of the labels in each batch with $y^*$, and leaving the remaining clean labels unperturbed.
>
> **Limitations**
>
> In the final version of the paper, we will expand the concluding section to include the limitations associated with the practicality of the proposed attacks (high frequency streams represent a challenge), and with the threat model (label attacks). We will also include the case of distributions with covariate shift and designing defence strategies as important directions for future work.
>
> We hope our answers have clarified the reviewer’s concerns and will provide grounds for a more favorable assessment of the paper; we would be happy to respond to further concerns during the discussion period.

---

> > ### Comment · Reviewer_61gD · 2023-08-21
> >
> > Dear Authors
> >
> > I would like to thank you for the efforts put into writing the rebuttal. However, several concerns of mine are still unresolved.
> > While I understand that the nature of this work is to establish baseline behavior's of online learners, it is still required to show the importance of this problem when the learner is somewhat smart. That is, one should consider *at least* simple defenses and show that such approaches fail in defending against the proposed attack.
> >
> > Second, since the problem setup tackles online learning, I am not sure about the validity of the results when considering stationary and small scale distributions such as CIFAR10 and MNIST. Hence, one should include distribution-changing datasets that are designed specifically for online learning (such as the one mentioned in the review) and study the problem there instead of small scale datasets.
> >
> > Third and regarding the unbounded computational budget: while I appreciate this extension, one should study the effect of making $\rho$ a function of how expensive the developed attack. Further, it is important to conduct the empirical experiments under this setting as well.
> >
> > That being said, I think that this work is not ready to be accepted and hence I will maintain my original score.

---

> > > ### Author Response · Authors · 2023-08-21
> > >
> > > We thank the Reviewer for the response. We are happy that our rebuttal addressed, at least partially, the Reviewer's concerns. We would kindly request the Reviewer to consider the following points before reaching their final decision:
> > >
> > > * **Simple defense mechanisms**. The design and effectiveness of defense strategies greatly rely on the specific attack scenario. For instance, in the context of federated learning, anomaly detection filters might be employed on weight updates by the central server. Conversely, in centralized learning, data may undergo scrutiny, such as feasible-set projection, to ensure label integrity (an example is to control that labels are integers, as discussed before). While we recognize the importance of addressing these instances, the study of defense mechanisms falls outside the scope of our current work. Our primary focus is on offering a broader understanding of label attacks on online learners, serving as a foundational framework for future explorations.
> > >
> > > * **Non-stationary datasets**. We appreciate the Reviewer's suggestion. We would like to point out that the experiments we carried out are very well aligned with the related literature [1, 2, 3, 4]. We will carefully consider non-stationary distributions as a possible direction for future work, as we acknowledge the relevance of the proposed analysis.
> > >
> > > * **$\rho$ experiments**. We would like to point out that, in our rebuttal, we did run empirical experiments using MNIST (see the attached file). Due to time constraints during the rebuttal phase, we restricted our analysis to this dataset and a simple logistic regression model. We are currently running the $\rho$ experiments for all empirical scenarios outlined in our draft, and we will include the results in the final version of the manuscript.
> > >
> > > Once again, we thank you for your valuable feedback.
> > >
> > > [1] Pang, T. et al. Accumulative Poisoning Attacks on Real-time Data (NeurIPS, 2021).
> > >
> > > [2] Lee, S. et al. Continual Learning in the Teacher-Student Setup: Impact of Task Similarity (ICML, 2021).
> > >
> > > [3] Zhang, X. et al. Online Data Poisoning Attacks (L4DC, 2020).
> > >
> > > [4] Wang, Y. and Chaudhuri, K. Data Poisoning Attacks against Online Learning (arXiv, 2018).

---

### Official Review · Reviewer_cwDs · 2023-07-19

**Soundness:** 3 good
**Presentation:** 2 fair
**Contribution:** 3 good
**Rating:** 6
**Confidence:** 2

**Summary:**

The authors conduct theoretical analysis and empirical evaluation of poisoning attacks in the online learning setting, where the attacker can intervene in the labels of sequentially provided data. As a result of the theoretical analysis, the authors show that the strength of the attack becomes discontinuously stronger as the batch size asymptotically approaches infinity. It is also shown that even a greedy attack can be as strong as that in the clairvoyance setting under an appropriate condition.

**Strengths:**

It introduces a new problem setup for poisoning, and although the analysis is performed in a very simple setting, it leads to interesting results.
It is interesting that a properly designed greedy attack is as effective as an attack in the clairvoyant setting.
Theoretical analysis utilizing optimal control theory is not very common, and it is useful to introduce such methods.

**Weaknesses:**

There is little discussion of threat models and attacker models when viewed as a security issue. There is no disagreement that poisoning in an online setting is a serious threat, but the paper does not discuss where the significance of considering this type of dirty label attack lies and whether it could be a significant risk. In particular, I wonder if there is any point in dealing with a setting where the batch size is infinite in an online setting.

**Questions:**

In equation 4, is a parameter to adjust the balance between g^per and g^nef necessary?
In equation 6, when d_mu(C,P)=1, it cannot be said that the student's prediction and the target's prediction coincide since they merely coincide in their expectations.

**Limitations:**

The subject is an attack on a learning system and it can affect society in a malicious way, but it is obvious that the research interest is in the theoretical aspect of the behavior of the system under attack.

---

> ### Author Rebuttal · Authors · 2023-08-09
>
> We thank the Reviewer for the constructive feedback. Please see below for our response.
>
> **Weaknesses**
>
> There is little discussion of threat models and attacker models when viewed as a security issue. There is no disagreement that poisoning in an online setting is a serious threat, but the paper does not discuss where the significance of considering this type of dirty label attack lies and whether it could be a significant risk.
>
> * There are several practical use cases that motivate our analysis. In federated learning, where data are collected independently by each node in the federation, malicious nodes could modify the labels before providing the model updates to the central server for aggregation. Label attacks can also occur in online learning settings where models are fine-tuned on the fly using user-generated data, such as email spam filters, product recommendation engines, chatbots, and fake review detectors. Note that, in streaming scenarios, the attacker can monitor the effect of previous poisoning on the learner and make adaptive decisions on how to poison next. This adaptability, inherent to the online setting, makes attacks on online learners potentially extremely dangerous. We will expand the introductory section of the manuscript to include these examples.
>
> In particular, I wonder if there is any point in dealing with a setting where the batch size is infinite in an online setting.
>
> * The Reviewer raises a good point: in practical contexts, it is unlikely that streaming data would arrive in batches of infinite size. We considered the case of infinitely large batches ($P\to\infty$) as an approximation for cases where $P$ is sufficiently large to make batch averages effectively equal to averages over the data distribution, which is a limit receiving much interest from the theoretical point of view recently [1-3]. This reduces the optimal control problem to a deterministic one, which we could solve exactly. We would like to point out that **our experiments approach the theoretical limit of $P\to\infty$ very rapidly**. See for example Fig. 2-B in the paper: for $P=10$, the steady state accuracy of the model already gets very close to the $P\to\infty$ prediction. Moreover, in practical applications, online learning may involve an accumulation phase where multiple batches are collected before being used to update the ML model, which represents a context where our analysis may apply.
>
> **Questions**
>
> In equation 4, is a parameter to adjust the balance between g^per and g^nef necessary?
>
> * This parameter is necessary to ensure that the two costs have the same scale, so that we can analyse the effect that the attack strength has on the steady state of the learner in a standardised fashion across architectures. In more detail, the nefarious cost, $g^{\mathrm{nef}}_{\mu}$, represents the distance between the learner and the target at training step $\mu$. This quantity can vary significantly depending on the setup of the experiment (the learner architecture, the target of the attacker, etc), thus affecting the amplitude of the perturbations. This makes it hard to compare results from different setups. In order to mitigate this effect, the perturbation cost, $g^{\mathrm{per}}$, is weighted by a factor that has the same magnitude of the average nefarious cost. We hope this clarifies the role of  $\mathcal{E}(\phi^*)$, and we are happy to answer any further question the Reviewer may have.
>
> In equation 6, when $d_{\mu}(C,P)=1$, it cannot be said that the student's prediction and the target's prediction coincide since they merely coincide in their expectations.
>
> * We thank the reviewer for pointing this out. We will make the suggested correction to the sentence.
>
> We hope that our answers have clarified the reviewer’s concerns and will provide grounds for a more favourable assessment of the paper; we would of course be happy to respond to further concerns during the discussion period.
>
> References
>
> [1] Ba et al. NeurIPS ‘22;
>
> [2] Damian et al. COLT ‘22;
>
> [3] Dandi et al. arXiv:2305.18270.

---

> > ### Comment · Reviewer_cwDs · 2023-08-22
> >
> > Dear Authors
> >
> > Thank you for your responses to the reviewer's questions and comments. After reading the responses, some of my questions are resolved. As the other reviewers pointed out, this work requires several clarifications reflecting the reviewers' comments for publication while I would like to keep the current score since I believe this manuscript contains an interesting approach to attacks on online learning.

---

> > > ### Author Response · Authors · 2023-08-22
> > >
> > > We thank the Reviewer for the positive evaluation of our submission. We are glad the Reviewer finds our work important for online adversarial attacks, and that we resolved part of the Reviewer's questions. Although the official discussion phase has concluded, we remain available to address any additional questions the Reviewer might have.
> > >
> > > We will of course incorporate in the final version of the manuscript all the clarifications that we provided in our rebuttal.
> > >
> > > Thank you again for your valuable feedback.

---

### Official Review · Reviewer_rUtu · 2023-07-21

**Soundness:** 3 good
**Presentation:** 3 good
**Contribution:** 2 fair
**Rating:** 6
**Confidence:** 3

**Summary:**

The paper provide an analysis of steady state of linear learners trained via SGD under online setting, where the found a phase transition in terms of attack strength. Some experiments are also conducted to demonstrate the insight from the theory.

**Strengths:**

The formulation of attacking learning model under online setting is clean and intuitive. The theory is also illuminating. The insight that model performance will sharply decrease if attach strength is higher than a threshold is an interesting observation. In addition, the theory is supported by experiments.

**Weaknesses:**

Overall my feeling is both empirically and theoretically, the paper is not strong. Theory side, the main concern I have on the work is that the problem might be oversimplified, thus it is not sure how much the insight can carry over to more realistic settings. On empirical side, there are quite a few simplifications in the setting analyzed, e.g., linear model, SGD only training. While I agree theory should start simple, it would be more convincing to see experiments on more realistic settings, i.e., to test if the insights from the simple model can hold  for more practical settings, even if they do not hold, it would be still interesting to see the results.

**Questions:**

1. If the training algorithm is not SGD, instead it is, say Adam, how would this change the analysis result conceptually? In addition, in experiment would it change anything empirically?

2. In Fig. 3, reinforcement learning and  clairvoyant methods are not compared for neural nets due to their complications. I can understand  clairvoyant might be infeasible due to high nonconvexity of the problem, but why RL methods are not feasible? It seems like the online adversarial attack could be a reinforcement learning problem by nature.

**Limitations:**

Limitation are sufficiently discussed.

---

> ### Author Rebuttal · Authors · 2023-08-09
>
> We thank the reviewer for the constructive criticism and for the enthusiastic description of the strengths of our paper.
>
> **Weaknesses**
>
> Overall my feeling is both empirically and theoretically, the paper is not strong. Theory side, the main concern I have on the work is that the problem might be oversimplified, thus it is not sure how much the insight can carry over to more realistic settings. On empirical side, there are quite a few simplifications in the setting analyzed, e.g., linear model, SGD only training.
>
> * To the best of our knowledge, this is the first paper to study online poisoning from the theoretical point of view. We therefore decided to start with the simplest possible setting, linear models, which have been used to study various deep learning phenomena recently (feed-forward neural networks [1], self-supervised learning [2,3], pre-training [4], implicit bias[5-6] etc.) This framework allows us to obtain exact and explicit results, which are easy to interpret and help us develop our intuition about the problem.
>
> While I agree theory should start simple, it would be more convincing to see experiments on more realistic settings, i.e., to test if the insights from the simple model can hold for more practical settings, even if they do not hold, it would be still interesting to see the results.
>
> * We agree with the reviewer that it is key to verify that the insights obtained theoretically need to be verified experimentally in more realistic settings. We point the reviewer to the experiments with deep networks (LeNet, VGG, ResNet) on MNIST and CIFAR10 (Sec. 4), which confirmed the key insights from our theoretical analysis carried over to these settings, in particular the existence of a critical threshold of attack strength beyond which catastrophic collapse in accuracy happens.
>
>
> **Questions**
>
> If the training algorithm is not SGD, instead it is, say Adam, how would this change the analysis result conceptually? In addition, in experiment would it change anything empirically?
>
> * Following the reviewer’s suggestion, we have run simulations for the case of logistic regression classifying MNIST data using Adam. The result is shown in the attached document - see Fig. A and B. The steady state reached by the learner is very similar to what we observe using SGD, thus providing further validation for our results. Due to the limited time allowed for the rebuttal, we could obtain the results for the logistic regression case only, and we will cover all real-data experiments considered in the draft for the final version of the paper. From a theoretical point of view, Adam has -- to the best of our knowledge -- defied an analysis even in the vanilla case so far (without attacks), with most recent work on learning dynamics focusing on vanilla SGD [6-9]
>
> In Fig. 3, reinforcement learning and clairvoyant methods are not compared for neural nets due to their complications. I can understand clairvoyant might be infeasible due to high nonconvexity of the problem, but why RL methods are not feasible? It seems like the online adversarial attack could be a reinforcement learning problem by nature.
>
> * The clairvoyant setting is not doable due to the high non-convexity of the problem, as the reviewer points out. Running the clairvoyant non-linear solver becomes infeasible for complex architectures: finding a solution to the equations takes a long time, and since the problem is non-convex, there are no guarantees the sequence of actions converges to the optimal one.
>
> * In the RL approach, the state space is given by the input data and the learner’s weights. The latter makes an RL approach prohibitively expensive for large architectures, if you train all the weights. Following the reviewer’s question, we considered an approach where the whole network is trained, but our TD3 agent only attacks the network using knowledge of the read-out weights and the inputs. The results are shown in the attached document, see Fig. C and D. The agent is able to perform better than constant attacks, though, in terms of running cost, it does not get as close to the greedy attacks as it does when it has full sight of the learner’s parameters (for a comparison, see the paper Fig. 3-A,B bottom panels, and the supplementary material Fig. 5-A,B bottom panels). We will add these evaluations to the final version of the paper, and a note to clarify why exactly clairvoyant and RL-based approaches become not practical for complex models.
>
> We hope that our answers have clarified the reviewer’s concerns and will provide grounds for a more favourable assessment of the paper; we would of course be happy to respond to further concerns during the discussion period.
>
> References
>
> [1] Saxe et al. ICLR ‘13;
>
> [2] Tian, Chen, Ganguli, ICML ‘21;
>
> [3] Jing, Vincent, LeCun, Tian, ICLR ‘22;
>
> [4] Wu, Zou, Braverman, Gu, Kakade, NeurIPS ‘22;
>
> [5] Moroshko, Woodworth, Gunasekar, Lee, Srebro, Soudry, NeurIPS ‘21; Li et al. NeurIPS ‘22;
>
> [6] Chizat & Bach NeurIPS ‘18;
>
> [7] Goldt et al. NeurIPS ‘19;
>
> [8] Ben Arous et al. JMLR ‘21;
>
> [9] Veiga et al. NeurIPS ‘22.

---

### Author Rebuttal · Authors · 2023-08-09

We would like to thank the reviewers for their constructive criticism. We are thrilled to see that the reviewers broadly agree this is an important, underexplored problem, and that our theoretical and empirical analysis offers some novel, non-trivial insights. We certainly agree that the scenario we study analytically is simplified, but we would like to point out that obtaining analytical results on learning dynamics is non-trivial even with simplified models, and the insights we derive are replicated in our analysis of deep networks (VGG, ResNet) on realistic data sets (MNIST, CIFAR10).

Concretely, the reviewers pose several interesting questions (detailed inline response below); while several such questions would necessitate a significantly broader scope than afforded by this paper, we have implemented a number of suggestions, namely:

* We have analysed empirically the dynamics of attacks when the learning algorithm employed is Adam, instead of vanilla SGD, observing again a very good agreement with our theoretical insights, see Fig. A, B (suggested by reviewer rUtu).

* We extended the reinforcement learning analysis to more complex models, under the proviso of using only the last layer weights to characterise the state of the student (unavoidable due to the very large state space resulting for using the full weight set of a DNN), with results in line with what we observed on simpler models, see Fig. C, D (addressing the questions of reviewers rUtu and 7DCo)

* We have examined analytically a simple defense mechanism/alternative scenario where only a fraction $\rho$ of labels are perturbed in each batch (as requested by reviewers 61gD and 7DCo). This can still be analysed theoretically in simple cases and reveals that the attack strength effectively decreases by a factor $\rho$.

* We will significantly expand our discussion of practical use-cases where our type of attacks can arise, namely the fine-tuning of systems learning from user-generated data and federated learning (suggested by reviewers cwDs, 61gD, and MYtD)

We hope that these extra analyses, and the detailed replies we give below, will be useful in informing a constructive discussion period, solidifying the positive outlook on the general work, and highlighting the necessity of some of the restrictive assumptions we make.

---

> ### Author Response · Authors · 2023-08-15
> **We are at the reviewers' disposal for any further questions**
>
> Since the discussion phase is about to end, we are writing to kindly ask if the reviewers have any additional comments regarding our response. We are at their disposal for any further questions. In addition, if our new experiments address the reviewers' concerns, we would like to kindly ask if the reviewers could reconsider their scores.

---

### Decision · Program_Chairs · 2023-09-21

**Decision:**

Accept (poster)

**Comment:**

This work studies the robustness of online learners to attackers that can perturb the data, specifically the labels, in order to inflict higher loss on the learner. The theoretical and empirical content of the submission consider the amount of data needed by the attacker, different strengths of attack, and different types of learner (from linear to small but real deep networks like ResNet-18). Five expert reviewers with backgrounds in adversarial training, attacks and poisoning, and continual learning are initially split between acceptance (cwDs: 6, 7DCo: 5, MYtD: 5) and rejection (61gD: 3, rUtu: 4). The authors provide a rebuttal, most reviewers (except MYtD, rUtu) engage in author-reviewer discussion, and rUtu engaged in AC-reviewer discussion. rUtu ultimately raised switch to accept (rating: 6) given the rebuttal and the results for different optimizers (SGD, Adam) and deep albeit toy architectures (LeNet, ResNet-18). Following all discussion, 4/5 reviewers side with acceptance save for the notable exception of 61gD, who objects to the problem statement (the threat model gives the attacker a lot of power in the streaming setting) and scope of the work (no "true" online problems like continual learning or dataset shift) as too limited. The AC sides with acceptance in agreement with the majority of reviewers: this work is theoretically and empirically informative, and the experiments are more practical than some more theoretical work that only addresses linear models, and publishing this work can guide more research on the topic.

To unpack the negatives in more detail: the argument for rejection is that the chosen setting is unrealistic and unmotivated (in a streaming setting, the attacker might not have access to labels, and the attacker may not have unbounded computation time because the environment/others might keep providing a stream of inputs to the learner). Discussion does not persuade 61gD, who requests the study of "true" online problems like continual learning or distribution shift instead of streaming stationary problems (like MNIST or CIFAR-10 classification). Furthermore, they want to see analysis and results with at least some form of defense. The AC agrees that these are all directions for progress in future work (and thanks 61Gd for the detailed and grounded review with references), but nevertheless sides with acceptance, so that this paper can contribute its findings for online learning and encourage more work in this area. This is especially true with the inclusion of the rebuttal material.

The AC encourages the authors to incorporate the feedback from reviews and in particular to discuss the limitations emphasized by 61gD. Clearly identifying the scope of what has been done and remains to be done is a service to the reader and can result in more impact rather than less.

Note: the AC acknowledges the comment from the authors regarding the initial lack of reviewer engagement. The AC pursued post-rebuttal discussion, resulting in eventual engagement by all but one reviewer, and considered each review-rebuttal point independently for the cases when the reviewer did not respond.